# Simultaneous representation of a spectrum of dynamically changing value estimates during decision making

David Meder [1,2], Nils Kolling[1,3], Lennart Verhagen [1], Marco K. Wittmann [1,3], Jacqueline Scholl[1], Kristoffer H. Madsen [2], Oliver J. Hulme[2], Timothy E.J. Behrens[3] & Matthew F.S. Rushworth[1,3]

Decisions are based on value expectations derived from experience. We show that dorsal anterior cingulate cortex and three other brain regions hold multiple representations of choice value based on different timescales of experience organized in terms of systematic gradients across the cortex. Some parts of each area represent value estimates based on recent reward experience while others represent value estimates based on experience over the longer term. The value estimates within these areas interact with one another according to their temporal scaling. Some aspects of the representations change dynamically as the environment changes. The spectrum of value estimates may act as a flexible selection mechanism for combining experience-derived value information with other aspects of value to allow flexible and adaptive decisions in changing environments.

---

[1] Wellcome Centre for Integrative Neuroimaging (WIN), Department of Experimental Psychology, University of Oxford, South Parks Road, Oxford OX1 3UD, UK. [2] Danish Research Centre for Magnetic Resonance, Centre for Functional and Diagnostic Imaging and Research, Copenhagen University Hospital Hvidovre, Hvidovre 2650, Denmark. [3] Wellcome Centre for Integrative Neuroimaging (WIN), Centre for Functional MRI of the Brain (FMRIB), Nuffield Department of Clinical Neurosciences, John Radcliffe Hospital, University of Oxford, Oxford OX3 9DU, UK. Correspondence and requests for materials should be addressed to D.M. (email: davidm@drcmr.dk)

When an organism makes a decision, it is guided by expectations about the values of potential choices. Estimates of value are, in turn, often dependent on past experience. How past experience should be used when deriving value estimates to guide decisions is not, however, always clear. While it might seem ideal to use the most experience possible, from both the recent and more distant past, this is only true if the environment is stable. In a changing environment it may be better to rely only on most recent experience because earlier experience is no longer informative[1,2].

Previous studies have focused on value learning: how value estimates are updated after the choice is made and the choice outcome is witnessed[1,2]. These studies have emphasized that each outcome has a greater impact on value estimates when the environment is changeable or volatile; the learning rate (LR) is higher and so value estimates are updated more after each choice outcome. Similarly, each outcome has a greater effect on activity in brain areas, such as dorsal anterior cingulate cortex (dACC) when the environment is volatile (Fig. 1).

However, while volatility affected dACC at the time of each decision-outcome, there was no evidence that it affected average dACC activity at the time of the next decision. It is therefore unclear how dACC activity might change as a function of the learning rate determining the choice value estimates that guide decision making at the point in time when decisions are actually made (Fig. 1). This is this question that we address here. Rather than investigating dACC activity at the time of decision outcomes and in relation to learning we focus instead on how dACC represents value estimates employed at the time of decision making.

When making decisions, the brain might first attempt to determine the best suited LR for the given environment and then consider a single-value estimate previously updated based only on this LR. If this is the case then there may be no overall change in average dACC activity but variance in dACC might best be explained by value estimates calculated at the best LR rather than other inappropriate LRs. Alternatively dACC might hold simultaneous representations of value estimates based on a broad spectrum of LRs. Although intuitively the former might seem computationally simpler, there is evidence that neurons in macaque dACC reflect recent reward experience with different time constants as might be expected if they were each employing a different LR[3–5]. However, the role of such neurons in behavior remains unclear. Here we sought evidence for the existence of value estimates in dACC and elsewhere in the human brain, based on experience over different timescales (and therefore employing different LRs), and examined how such representations mediate decision making (Fig. 1).

We developed a new approach to analyze neural data going beyond the typical use of computational models in investigation of brain behavior relationships. Typically, the free parameters of a computational model (e.g., LR) are fitted to the behavior of the subject from which trial-wise estimates of the computed variables can be extracted (e.g., value estimates). However, here we also test whether neuronal populations exist with responses that are better characterised by parts of parameter space that are not overtly expressed in current behavior. Identification of such representations is precluded by focusing exclusively on the parameters currently expressed in behavior. Previous investigations have considered neural correlates of model parameters fit to models that do not correspond to the current behavior e.g., refs [6,7] and the issue of the similarity of neural correlates of models with different parameterizations[8]. However, here we aim to reveal the dynamic changes and topography of "hidden" information by fitting LR values to each voxel independently, visualizing those parameters over anatomical space and computing their interactions. Instead of investigating where in the brain clusters of voxels express similar neural activity related to value estimates, here we examine the range of value estimates across voxels. We also examine changes to this pattern as a function of volatility.

## Results

**Experimental strategy**. We used fMRI data from 17 subjects acquired during a probabilistic reversal learning task[1]. Subjects repeatedly chose between two stimuli with visible reward magnitudes and hidden reward probabilities that had to be learned through feedback (Fig. 2a). Thus in this experiment subjects had to use past experience to estimate reward probabilities for each choice. Accordingly, reward magnitude estimates should be based on the stimuli displayed on each trial but the reward probability estimates should depend on recent experience over several trials. The reward probability might be estimated with different LRs depending on how quickly the environment is changing[1]. Each choice's value can then be derived by combining the explicit reward magnitude with the estimated probability of receiving the reward. Each session comprised two sub-sessions (order counterbalanced across subjects): one where reward probabilities remained stable and another sub-session where reward probabilities were volatile (Fig. 2b). The transition between the two sub-sessions was not announced to the subject.

In order to investigate whether the human brain represents multiple reward probability estimates that are based on a spectrum of LRs, we used a novel approach to analyze fMRI data. In addition to other regressors modeling standard variables of interest (such as the reward magnitudes displayed to subjects on the screen, the reward received, and so on) and physiological noise, we added two regressors, one modeling the estimated

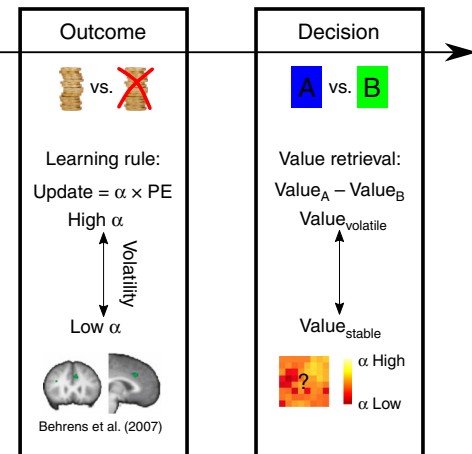

**Fig. 1** Outcome, choice, and learning rate. When outcomes of decisions are witnessed, the prediction for the next choice is updated based on a learning rule where the prediction error (PE) is weighted by the learning rate α. Behrens et al. have shown that average activity in dACC reflects the environment's volatility and that under high volatility, the options' values are updated with a high learning rate α. However, at the time of the actual decision on the next trial, volatility no longer exerts a significant effect on average dACC activity. However, the representation of choice value estimates necessary for decision making (the value estimate for option A relative to that of option B) might be represented in some other way such as an anatomically distributed pattern of activity where different value estimates might be calculated with different parameterizations of α, depending on the volatility. Copyright for brain image: Behrens et al., Learning the value of information in an uncertain world, 2007, Nature Neuroscience, all rights reserved

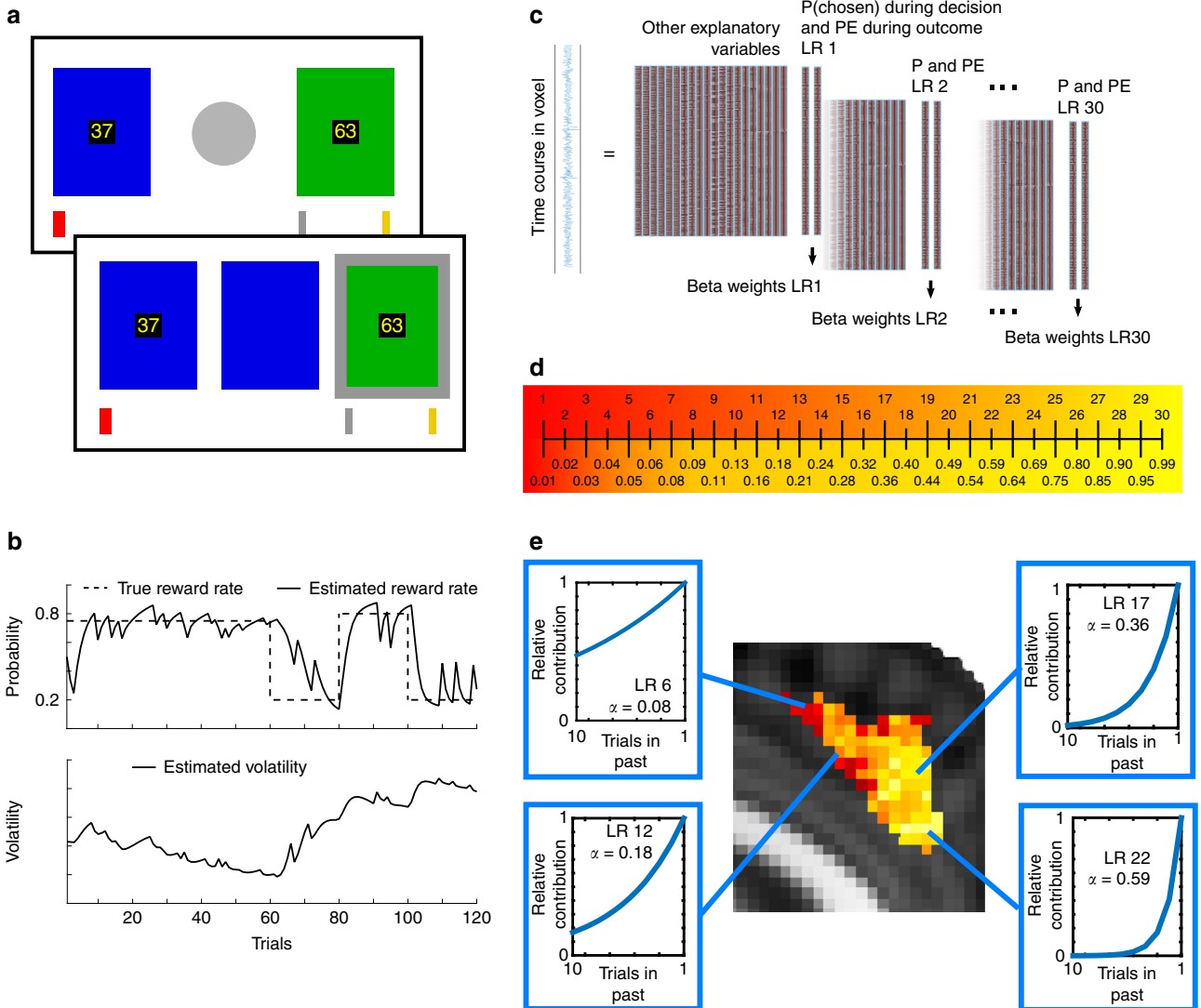

**Fig. 2** Methods and analysis. **a** Probabilistic reversal learning task. Subjects chose between a green and a blue stimulus with different reward magnitudes (displayed inside each stimulus). In addition to the random reward magnitude, stimulus value was determined by the probability of reward associated with each stimulus which drifted during the course of the experiment and had to be learned from feedback. After the choice was made, the red bar moved from left to right if the chosen option was rewarded. Reaching the silver bar was rewarded with £10, the gold bar with £20. In this example situation, the subject had chosen the green stimulus (gray frame), but was not rewarded so the red bar did not move. **b** Example of reward probability schedule and estimated volatility of the reward probability from a Bayesian learner when the stable phase came first[1]. Each session had a stable phase of 60 trials where one stimulus was rewarded 75% of trials, the other 25%, and a volatile phase with reward probabilities of 80 vs. 20%, swapping every 20 trials. The order was counterbalanced between subjects. Note, the reward rate and volatility estimates from the Bayesian learner are only shown to convey task structure and the difference in volatility between sub-sessions. The Bayesian learner model was not used for analysis. **c** Analysis. As in a conventional fMRI analysis, the blood-oxygen-level-dependent (BOLD) signal time course in every voxel was analyzed in a GLM with a design matrix containing relevant regressors. Additionally, one of the regressors modeled the estimated reward probability of the chosen option during the decision phase, another one the prediction error during the outcome phase. The same LRs were used for deriving the reward probability estimates and the prediction error regressors (these two regressors are referred to collectively as LR regressors). This analysis was repeated 30 times, deriving the beta-values for probability estimates and prediction errors based on 30 different LRs, testing their ability to explain signal variance. **d** With equally spaced LRs across the LR spectrum (0.01–0.99) the regressors would be more strongly correlated at higher LRs, therefore we derived 30 LRs with larger intervals between higher LRs, resulting in uniform correlation across the spectrum. **e** In a highly volatile environment, the stimulus-reward history should be more steeply discounted (higher LR) because information from many trials ago is likely to be outdated. The blue decay functions show the relative contribution of the previous trials' outcomes to the current reward probability estimation with different LRs. We derived the best-fitting LR for every voxel in every subject. For example, within dACC the BOLD signal in some voxels is best explained by a low LR (red), in others by a high LR (yellow)

reward probability of the chosen option during the decision phase, another one modeling the prediction error during the outcome phase. We repeated this entire analysis 30 times for probability estimates and prediction errors based on 30 different LRs ranging from 0.01 to 0.99 (slow to fast LRs), deriving the best-fitting LR for every voxel (Fig. 2c–e). In other words, the 30

repetitions of the analysis make it possible to derive 30 different estimates of the reward probability based on 30 different LRs. The 30 different LRs were chosen so as to sample the entire LR space between 0.01 (almost no learning) and 0.99 (almost complete revision of value estimates on each trial) and to be equally spaced in terms of their correlation to the neighboring regressors

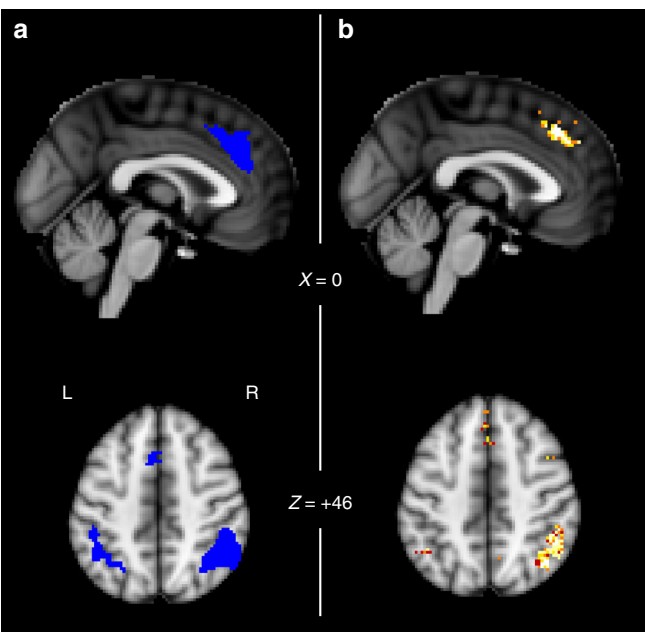

**Fig. 3** Regions of interest. **a** dACC and IPL regions defined by conjunction of (1) anatomical masks for dACC and IPL from the connectivity-based parcellation atlases (http://www.rbmars.dds.nl/CBPatlases.htm)[19,20] and (2) significantly decreasing activity (blue) associated with the magnitude of the chosen option during decision. **b** The dACC and IPL regions showed high evidence for coding LRs (posterior exceedance probability >0.95)

(Fig. 2d; Methods section). In their previous study Behrens et al.[1] assumed one dynamic, but unitary LR generating value estimates across the brain. However, in this study we instead compared value estimates generated by 30 stable learning rates. Thus, assigning a best-fitting LR to each voxel based on its own data reveals a pattern of diverse value estimates based on different time periods of experience (different LRs). The best-fitting LR of a voxel corresponds to the value regressor calculated with an LR that explained most of the variance in the voxel's time course, compared to the other LR regressors, regardless of how much variance it actually explains. While such an approach is unlikely to capture the full range of factors affecting activity in a voxel it has the potential to identify relationships between brain activity and choice value estimates that cannot be captured with standard analysis techniques.

We combined two approaches to define the brain areas that we investigated in detail. First, a priori we anatomically defined two regions of interest (ROIs) that are likely to be important for using feedback to adapt and change decision making. First, we considered the dACC because of its role both in monitoring feedback and decision outcomes and in adaptive control of subsequent behavior and because its outcome-related activity is known to be related to the learning rate dACC[1,9–16]. It is also the area in which activity has been most consistently related to adjustment in the speed of behavioral change[1,17]. In addition we considered the inferior parietal lobule (IPL) because it is frequently co-activated with the dACC during decision making tasks[18]. The anatomical masks for dACC and IPL were taken from connectivity-based parcellation atlases[19,20]. Subsequently, we checked that these regions were task-relevant by looking for activity that was associated significantly with the expected reward magnitude of the choice taken. This provides an orthogonal contrast to identify regions in which activity might also be related to expected reward probability estimated over different time-scales. Our ROIs were made from the conjunction of the anatomical and reward magnitude task-relevant activity (Fig. 3a).

We also used a second independent approach to identify ROIs. This approach identified very similar areas in dACC and IPL (Fig. 3b) and in two other brain regions (Supplementary Fig. 1). The analysis approach that we have described so far (Fig. 2) assigns a best-fitting LR in comparison to all other LR regressors, but it does not quantify whether this best-fitting regressor explains any significant amount of evidence. In order to confirm that the voxels in our ROIs reflected activity that was related to probability estimates in a manner that could not be due to the overfitting of any particular LR in a given voxel, we ran a singular value decomposition (SVD) over the LR regressors (before HRF-convolution, normalization and high-pass filtering) to derive singular values capturing most of the variance associated with the LR regressors. The first three singular values explained on average 99.63% of variance in the LR regressors and can thus be interpreted as capturing almost all LR-related variance. A standard t-test over the parameters of these three regressors in a GLM explaining the measured BOLD signal in each voxel is not an appropriate test because it assumes a coherent positive or negative correlation between the regressor and the measured BOLD signal in a given voxel across all subjects. An F-test over the parameters might therefore be appropriate but there is currently no widely agreed standard approach for combining individual subject F-test results into a group-level analysis in neuroimaging. We therefore used a different approach to test whether a voxel's time course reflected LR-related information. For every voxel, we derived the Akaike Information Criterion (AIC) scores from our main GLM (the seven regressors of no interest and their temporal derivatives (Methods section) plus six motion confound regressors, but in the absence of any LR regressors). This reveals how well a model lacking multiple LRs accounts for activity variation in every voxel in the brain. We also ran an identical GLM that contained the same regressors but also the first three principle components from the SVD (HRF-convolved, demeaned and high-pass filtered), and again computed the AIC score. This reveals how well a model containing LR-based reward probability estimates accounts for activity variation in every voxel in the brain. We then compared the AIC scores of the two models of brain activity at every voxel using random-effects Bayesian model comparison for group studies[21]. This procedure returned protected exceedance probabilities for every voxel, revealing the degree to which the model containing the singular values, reflecting value estimates based on one or multiple LRs, was the more likely model of the neural data (Fig. 3b). The protected exceedance probability can be considered a quantitative measure of the evidence for the model with LR-related information. The random-effects Bayesian model comparison tests for the amount of evidence in favor of a model across all subjects, regardless of sign. Since exceedance probabilities of all compared models sum to 1, they can be easily interpreted in a way that is similar to classical p-values[21,22]. In voxels with a protected exceedance probability of >95%, this corresponds to a 95% confidence that the model with singular value regressors is the most likely model to explain the activity. Thus, we can state that in these voxels LRs have an impact on activity. Having established initial candidate areas of interest in an unbiased way we then went on in subsequent analyses to establish more specifically how reward probability estimates based on different LRs were represented.

The relevance of the dACC and IPL regions that we had defined a priori based on anatomy was confirmed: these ROIs showed high evidence of coding reward probability estimates based on LRs. Accordingly, for subsequent analyses, we constrained the ROI masks to those voxels that fulfilled both the anatomical and task-relevant exceedance probability criteria. We found two further clusters with high evidence in the right

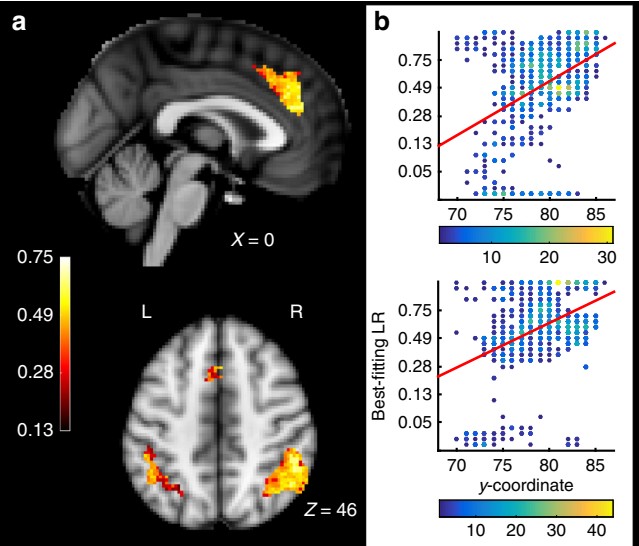

**Fig. 4** Topographic maps of LRs. **a** A topography of diverse estimates of the reward probability based on different LRs exists in the ROIs. Bright yellow and white colors indicate voxels with high LRs while darker, redder voxels indicate voxels with lower LRs. The color bar on the left indicates the set of LRs (high LRs at top, low LRs at bottom) chosen in 30 steps to minimize correlation between regressors in LR space (see also Fig. 2d). **b**. Spatial gradient along the rostrocaudal axis in dACC in two example subjects. Each voxel's best-fitting LR is plotted against its position on the y-coordinate. Note that the color of the dots reflects the number of voxels having a given combination of values (see color bars beneath graph). Red lines: regression of all voxels' best-fitting LR against their y-coordinate

frontal operculum (rFO) and bilateral lateral frontopolar cortex (FPl) (Supplementary Fig. 1A). We focus on reporting results for our primary regions of interest, dACC and IPL, but in the Supplementary Information, we show related results for rFO and FPl. Using a different model, with an additional regressor coding the outcome of the trial (win or loss), the evidence in favor of an LR-based model in these regions was even stronger (Supplementary Note 1, Supplementary Fig. 1B). Furthermore, we have run two complementary analyses comparing the model with singular values first against a model with choice values derived from the optimal (gain-maximizing) LR, and second against a model with the choice values derived from the LR fit to subject behavior, with a similar result (Supplementary Note 1, Supplementary Fig. 2). Both these analyses support the finding that the ACC and IPL show evidence for coding multiple LRs. This finding is consistent with several other demonstrations that value representations in dACC often in tandem with IPL and FPl, guide stay/switch or engage/explore decisions of the sort that might be used to perform the current task in humans[12,23–27] and other primates[28,29]. In a later section below, we describe how such activity is not found in other brain regions also known to carry value representations.

**Diversity and topography of value representation**. The high exceedance probabilities in dACC and IPL reveal that LRs in a very general sense have an impact on activity in these regions; activity in these regions is better explained when learning rates are considered. However, what the analysis does not address is whether different voxels represent probability estimates based on different LRs and whether there is any topographic structure in such a representation. We turn to this question next. Using our multivariate mapping approach, we found that in our ROIs,

voxels did not homogeneously integrate the reward history with the same LR, but that there was some degree of spatial topographic organization of the diverse probability estimates (Fig. 4). First, we asked whether the spatial distribution of the best-fitting LRs was entirely random or whether it was organized in three-dimensional space, we tested whether the voxels' x, y, and z-coordinates (Montreal Neurological Institute (MNI) space) could predict the best-fitting LR. While these axes do not correspond exactly to anatomical features such as gyri or sulci in these regions, this analysis is a first indication of spatial organization. In both IPL and dACC, a significant amount of variability in the best-fitting LRs in voxels was explained by the x, y, and z-coordinates of the voxel when regression models were fitted to each subject's data (t-test over the variance explained by every subject's regression model ($r^2$) against the mean $r^2$ of 10,000 regression models with randomly permuted coordinates. dACC: Mean $r^2$ true data = 0.101, mean $r^2$ permuted data = 0.002, $t_{16}$ = 5.071, $p < 0.001$, IPL right hemisphere: mean $r^2$ true data = 0.124, mean $r^2$ permuted data = 0.003, $t_{16}$ = 5.566, $p < 0.001$, IPL left hemisphere: Mean $r^2$ true data = 0.182, mean $r^2$ permuted data = 0.006, $t_{16}$ = 5.040, $p < 0.001$). The principle axis of anatomical organization in dACC in humans and other primates is approximately rostrocaudally oriented[20,30]. Although this axis does not fully correspond to the cardinal axes in the standard space for illustrating neuroimaging data (Montreal Neurological Institute (MNI) space) we nevertheless examined whether LRs were also organized along the MNI y-axis. Consistently, across subjects, in the dACC, LRs showed a gradient along the MNI y-axis with increasing LRs in the rostral direction (t-test of subjects' regression coefficients of the y-coordinate regressor against 0, $t_{16}$ = 2.175, $p = 0.045$, Supplementary Fig. 3). No major direction of anatomical organization has previously been reported for the IPL.

Previous studies have suggested that some brain regions may reflect a particular timescale of experience or LR that is appropriate to its function[31] but our analysis suggests dACC and IPL are, in addition, representing a spectrum of different LRs. Other relatively abstract features, such as numerosity are known to be represented topographically even though such representations do not map onto sensory receptors or motor effectors in any simple manner[32]. The distribution of LRs in dACC might approximately be related to the rostral-to-caudal gradient in its connectivity with limbic vs. motor areas[33].

**Mechanisms of adaption to changes in the environment**. As already explained, in a volatile environment, ideally decisions should be based on probability estimates derived from voxels with higher LRs, while in a stable environment, voxels with lower LRs might inform the decision. This suggests that one of two changes to the representation might occur as volatility of the reward environment changed. First, voxels might have dynamically changing LRs, depending on the environment (Fig. 5a). Alternatively, each voxel might retain its best-fitting LR regardless of volatility but the degree to which variance in each voxel's activity was explained by reward probability estimates with the best-fitting LR might get stronger in high LR voxels in volatile environment (or stronger in low LR voxels in stable environments). In other words, the regressor effect size (beta-weight) in high LR and low LR voxels might increase and decrease in volatile and stable environments, respectively (Fig. 5b). To probe these hypotheses, we split the BOLD signal time course into stable and volatile sub-sessions and again identified the best-fitting LR for every voxel in each of the two sub-sessions. We then compared the best-fitting LR in each sub-session in every voxel.

In the dACC and IPL, the LRs of the voxels' probability estimates were approximately normally distributed (Lilliefors test:

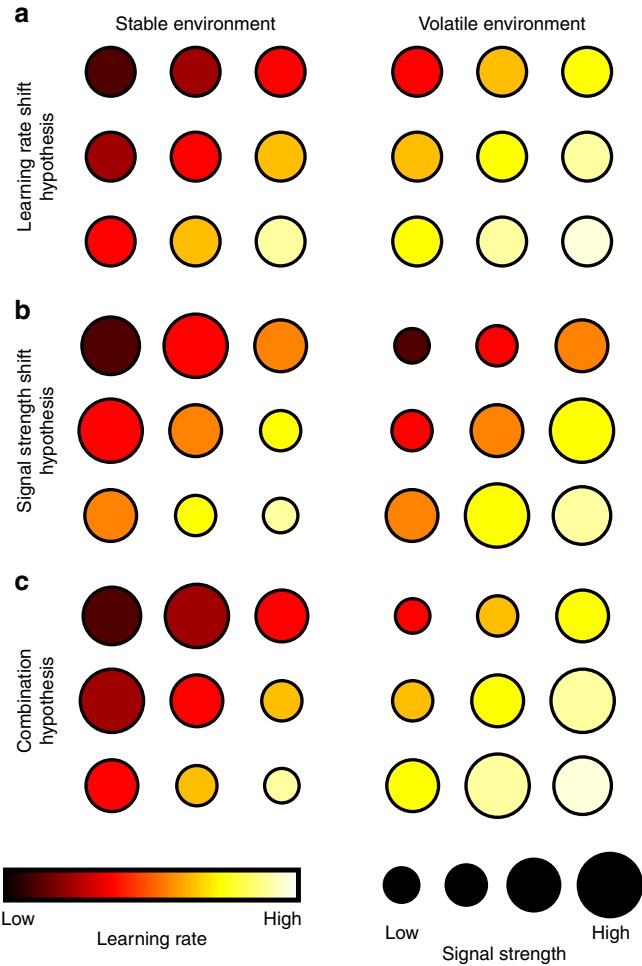

**Fig. 5** Schematic figure depicting possible ways in which multiple value estimates, based on different periods of experience determined by different LRs, might be represented in the brain as indexed by fMRI. We consider how such representations might change as the environment's volatility changes. Each row shows the representation of value estimates in nine example voxels in a stable and in a volatile environment. **a** According to the LR shift hypothesis, in a stable environment neurons in more voxels would compute value estimates based on lower LRs while they would shift toward higher LRs in a volatile environment. **b** The signal strength shift hypothesis predicts that the value estimates computed by the neurons of each voxel remain constant in all environments, but that those voxels with value estimates that are currently most relevant for the environment (high LR voxels in volatile environments and low LR voxels in stable environments) increase their signal strength. **c** The combination hypothesis suggests a combination of the two mechanisms in **a**, **b**

dACC $p = 0.363$; IPL $p = 0.950$) but they had significantly higher LRs in the volatile compared to the stable sub-session (average LR difference in dACC: 5.36 (details of LR scaling are shown in Fig. 2d), $t$-test of each subject's mean change in LR's against 0: $t_{16} = 3.68$, $p = 0.002$, average LR difference in IPL: 4.34, $t_{16} = 2.58$, $p = 0.020$) (Fig. 6). This finding suggests an adaptation mechanism resembling the one outlined in the shift-hypothesis (Fig. 5a). However, there might also be a change in how much of the neural activity in a voxel can be explained by the best-fitting LR. This would constitute a change in the effect size or beta-weight of the best-fitting regressor (Fig. 5b, c).

We therefore tested whether there was a dynamic change in the effect sizes of the best-fitting LRs depending on which LRs were currently behaviorally relevant. If such a boosting of relevant LR

signals exists, then we would expect voxels with lower best-fitting LRs to have higher beta-weights in the stable sub-session (a negative correlation between best-fitting LR and beta-weight) and voxels with higher best-fitting LRs to have the higher beta-values in the volatile session (positive correlation between best-fitting LR and beta-weight). We calculated the correlation between best-fitting LR and beta-weights for every subject in the two sub-sessions and transformed the correlation coefficients to $z$-scores (Fisher transformation). In the dACC, there was indeed such a dynamic change in effect size (mean difference in $z$-scores stable minus volatile sub-session $-0.230$, $t_{16} = -3.802$, $p = 0.002$), while this was not the case for the IPL (mean difference $-0.056$, $t_{16} = -0.818$, $p = 0.425$.) (Fig. 7). This shows that in the dACC, there is a combined adaptation of both the best-fitting LRs in voxels and a change in the effect size of the best-fitting LR, depending on the behavioral relevance of the best-fitting LR in a given environment (Fig. 5c). Thus, voxels change so as to code LRs appropriate for the current environment and they change so as to encode appropriate LRs more strongly than inappropriate LRs. In the IPL, however, only the former adaptation to the environment seems to take place (Fig. 5a).

**LRs as an organizational principle of interregional interaction.** So far we have seen that four brain regions carry multiple estimates of the value of choices that are based on different time constants of experience corresponding to different LRs. Thus, multiple LRs constitute an organizing principle determining distribution of activity patterns within these areas. We therefore next asked whether multiple LRs exerted a similar influence over the manner in which the areas interacted with one another. In other words, do voxels that code recent reward probability experience with a small time constant (high LR) in one brain region (e.g., dACC) interact preferentially with voxels with high LRs elsewhere? Similarly, are low LR voxels in different brain areas preferentially interacting with one another?

For every subject, we extracted the mean residual BOLD time course for all voxels after regressing out all the information contained in our original design matrix (coding, for example, for the various task events) and additionally all 30 LR regressors indexing the estimated reward probability in the decision phase and all 30 LR regressors indexing prediction error in the outcome phase. Thus, the residual time course no longer contained any LR related information. We then created a mean residual time course for all voxels originally identified as being of the same LR within each ROI and correlated these 30 mean residual time courses with the 30 mean residual time courses of another region. We found that the more similar the best-fitting LRs, the higher was the correlation of these voxels' residual time courses between the dACC and the IPL, as reflected in higher average correlation values along the diagonal (Fig. 8). For example, voxels with high LRs in the dACC were more correlated with high-LR voxels compared to low-LR voxels in the IPL (Fig. 8; bright yellow diagonal line running from top left to bottom right).

The statistical test for demonstrating the significance of the effect is best understood with reference to Fig. 8. It is to examine whether the subjects' $z$-transformed correlation coefficients are correlated positively with their closeness to the diagonal; this was indeed the case (negative Euclidian distance, one-tailed $t$-test of $z$-transformed correlation values $t_{16} = -2.944$, $p = 0.005$); the correlation between the brain areas' signals became greater the more that the signals were drawn from voxels with similar LRs.

In summary, even after removing all linear task-related information (activity linearly related to task variables and value estimates), voxels with the same best-fitting LR shared a more similar pattern of activity in dACC and IPL. Thus, LRs are not

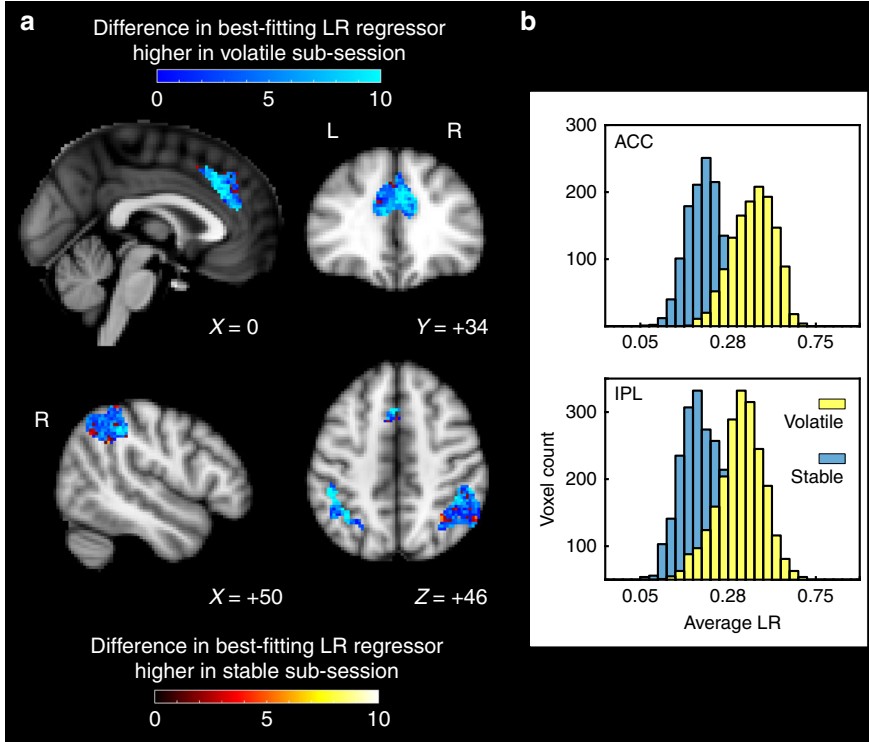

**Fig. 6** Dynamic changes in LR between stable and volatile sub-session. **a** Change in LR in every voxel between stable and volatile sub-session. Values on the color bars show the change in LR rank. **b** Distribution of number of voxels with best-fitting LRs in the two regions of interest

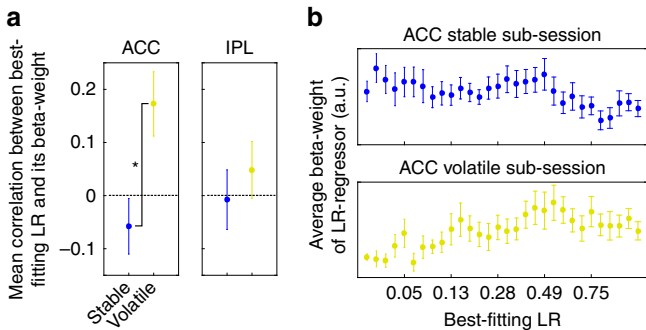

**Fig. 7** Change in the correlation between beta-weights of the best-fitting LR regressors and the best-fitting LR between sub-sessions. **a** In the dACC, the correlation was significantly positive for the volatile sub-session and significantly different from the negative correlation seen in the stable phase. **b** Average beta-weights across the whole spectrum of LRs in stable and volatile sub-session in the dACC. Error bars: Standard error of the mean

just an organizational feature of individual brain regions but also an organizing principle determining how these regions interact with one another. This feature of interactions between areas was also apparent in all combinations of interactions between all the four regions that showed high evidence for the coding of reward probabilities based on multiple LRs (ACC, IPL, FPl and rFO; Supplementary Fig. 7, Supplementary Table 1).

**Ubiquity of dynamic topographic value representations.** We have presented evidence for topographic organization of value estimates as a function of different LRs and shown LRs are an organizational principle of connectivity between regions, such as dACC and IPL. We next asked whether such representations and

interaction patterns are ubiquitous in all brain areas signaling value. We therefore performed the same analyses in another brain region that has repeatedly been linked to value and decision making, the ventromedial prefrontal cortex (vmPFC)[12,34–41]. In most studies, the strongest value-related activation was found in the anterior part of the vmPFC. We examined two vmPFC regions: anterior vmPFC and posterior vmPFC (Supplementary Note 2). We found some, albeit weak, evidence for LR related activity in anterior vmPFC (Supplementary Fig. 1C). Unlike in dACC and IPL, in vmPFC the amount of BOLD variance explained by SVD-derived singular values reflecting the LR regressors was not significantly greater than the amount of variance explained by a model lacking LR information (mean protected exceedance probability in anterior vmPFC = 0.478, $t$-test against 0.5: $t_{1007} = -5.19$, $p < 0.001$. Mean protected exceedance probability in posterior vmPFC = 0.340, $t_{1241} = -33.274$, $p < 0.001$). In fact, when the same statistical approaches were used as in our investigation of dACC and IPL we found that activity in many voxels in vmPFC was better explained by a model lacking the LR regressors. Value estimates with different LRs could be fit to voxels in vmPFC (Supplementary Note 3, Supplementary Fig. 4) but there was no shift in the distribution of LRs depending on the volatility of the environment (Supplementary Fig. 5, compare to Fig. 6) and there was no change in the correlation between the best-fitting LR and its beta-weight as seen in the dACC (Supplementary Fig. 6, compare to Fig. 7) in either vmPFC region. Additionally, unlike dACC, IPL, rFO, and FPl, there was no evidence that voxels in either vmPFC region preferentially interacted with voxels with similar LRs in other brain regions (i.e., no diagonal with high correlation values; Supplementary Note 4; Supplementary Fig. 7, Supplementary Table 1, compared to Fig. 8). In general, the average correlation over all voxels between two regions was significantly higher for dACC, IPL, rFO, and FPl than between any of these areas and either vmPFC subdivision (Supplementary Table 2).

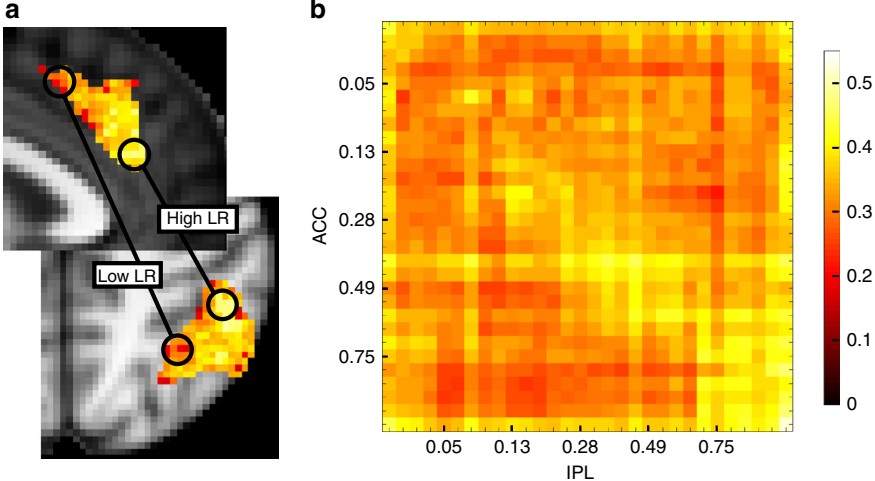

**Fig. 8** LR topography as an organizing principle for interaction between regions. **a** We investigated whether voxels that represent choice values with similar LRs also show stronger connectivity between regions. **b** Correlation plot depicting the correlation of the residual BOLD time course averaged over all voxels with the same best-fitting LR within dACC with the residual BOLD time course over all voxels with the same best-fitting LR within IPL, averaged over all subjects. The subjects' z-transformed correlation coefficients were correlated positively with their closeness to the diagonal

In summary, there is only comparatively weak evidence for the vmPFC holding value related information that reflects recent experience of reward probability and the value estimates it held were not as sensitive to environmental volatility. Thus, the neuroanatomical gradients of probability estimates calculated with different LRs in dACC and IPL, their sensitivity to environmental volatility, and their interregional LR-specific connectivity are not ubiquitous features of all value encoding brain regions. This supports the notion that the spectrum of value estimates based on multiple LRs that we find in some brain regions cannot be attributed to noise over subjects, time, or voxels.

**LR-based representation at decision outcome.** Finally, while the current investigation is focussed on the decision-making process, rather than the outcome monitoring phase of the task, we wanted to know whether we could observe comparable dynamic adaptations to environmental volatility during the outcome phase. We therefore investigated whether prediction error coding in ventral striatum (VS) would also reflect adaptations of which LRs should be expressed as a function of volatility. A model containing the first three singular values from an SVD over the prediction error regressors provided a good model of right VS activity during the outcome phase of the trials (Supplementary Fig. 8A). However, using a bilateral anatomical mask of the VS (Automated Anatomical Labeling (AAL) atlas[42]), the distributions of the LRs generating the prediction error were stable and did not change between the stable and volatile sub-sessions (Supplementary Note 5; Supplementary Fig. 8B). As in the study by Behrens et al.[1], in the current study there was an overall change in dACC activity during outcome but no evidence for a prediction error signal in dACC, using either standard analysis procedure similar to those used before[1] nor based on Bayesian group model comparisons such as those employed here.

**Discussion**
A number of cortical regions have been implicated in reward-guided decision making and it is possible that they operate partly in parallel[15,35,43]. For example, some aspects of decision making behavior are predicted by activity in vmPFC while others, even in the same task and at the same time, are better predicted by activity in the intraparietal sulcus[35].

DACC may be particularly important when deciding whether to switch and change between choices and behavioral strategies[12,13,15,23–29]. A flexible behavioral repertoire would be promoted by having multiple experience dependent value estimates, estimated over different timescales: representations of how well things have been recently and, simultaneously, how well they have been over the longer term. By contrasting the strength of such representations a decision-maker would be able to know whether the value of their environment is stable or improving or whether it is declining and that it might be time to explore elsewhere[27].

In the present study, we have found evidence that indeed multiple value representations, with different time constants, are especially prominent in dACC and IPL. A diversity of value estimates based on a spectrum of LRs could either reflect features of the neural representation guiding decision making, or it might simply be a reflection of natural variability over samples, trials, and voxels. Several aspects of our findings suggest that they reflect features of neural activity rather than noise. First, multiple LR-based representations were not ubiquitous; they were prominent in only a subset of regions implicated in value representation and decision making (Supplementary Notes 2–3, Supplementary Figs. 1–7). Second, the multiple LR representations were structured; they were topographically organized within areas (Fig. 4) and they were an organizing feature of interaction patterns between areas (Fig. 8). The conclusion that there are multiple LR-based value estimates is derived from averaging data over trials; in the future it might be interesting to examine the nature of these representations on a trial-by-trial basis.

While the parallel information processing entailed by such a representation might appear an unnecessary waste of computational resources, it may be advantageous when the volatility of the environment is changing and other LRs generate better value estimates than the one currently employed to guide behavior. Imagine a decision-maker that has estimated that the current environment is volatile and estimates choice values only on the basis of recent experience (high LR). If the decision-maker realizes that actually the environment is more stable than suspected, then it needs to retrieve the outcomes of earlier decisions and reweigh each of them according to the LR that is now optimal for estimating choice values. Our evidence suggests that the brain may compute many values estimates in parallel over different timescales and that such longer term timescale estimates (lower

LR estimates) are immediately available for the decision-maker to switch to on realizing the true level of environmental volatility. Since these value estimates are derived in a Markov decision process, only the most recent value estimate has to be remembered and updated so that it is not necessary to remember preceding outcomes. We are aware of one other study showing the representation of multiple estimates for latent decision-variables, in that case of reward prediction errors based on different discount factors[7]. However, here we tried to fully exploit the potential of topographic representations by showing their dynamic adaptations, as well as interregional connectivity patterns.

The co-existence of multiple experience dependent value estimates guiding decisions is also consistent with the results of single unit recordings made in macaques[3] in a dACC region homologous with the one we investigated here[20]. Neurons that varied in the degree to which their activity reflected just recent outcomes or also outcomes in the more distant past were also reported in the intraparietal sulcus and dorsolateral prefrontal cortex[3]. In the present study we also found evidence for such response patterns in fMRI activity in an adjacent part of the parietal cortex (IPL), a very rostral part of prefrontal cortex (FPl), and in FO. By recording activity in individual neurons it is possible to demonstrate precisely how different neurons, even closely situated ones, can code both recent and more distant rewards with different weights. In our study, however, by manipulating the reward environment that subjects experienced in volatile and stable sub-sessions, it was possible to show how such experience dependent reward representations changed with environment and behavior.

The evidence for value learning using multiple LRs in several cortical areas fits well with the idea that there exists a hierarchy of information accumulation from short timescales in sensory areas to long timescales in prefrontal, dACC, and parietal association areas[44–49]. In reinforcement learning, information obtained many trials ago in the past can still influence probability estimates when LRs are low. In our task, with an average trial duration of 20 s[1], information from several minutes ago has to be remembered. However, we can also show that even within a single area, there are gradients of timescale representation and that these representations are not fixed, but dynamically responding to the environment.

In situations in which dACC value representations guide behavior there are often also value-related activations in FPl and IPL[13,14,34,50,51]. Typically, these areas differ from others such as vmPFC in that they encode the value of behavioral change and exploration. In addition, in the present experiment we were able to show that there are links between the value representations in dACC and other brain regions. This suggests that multiple value representations of recent experience constitute an organizing feature of inter-areal interaction. It is not just that average activity throughout one region is related to the average activity of another. Instead parts of dACC employing the fastest and slowest LRs are interacting with corresponding subdivisions of FPl, IPL, and rOP. The pattern of results is suggestive of a distributed representation across multiple brain regions in which the value of initiating and changing behavior is evaluated over multiple timescales simultaneously[52].

In a longer behavioral testing session (without fMRI acquisition) it was shown that subjects do adapt their LR in response to changes in the volatility of the environment[1]. The change in best-fitting LRs that we observe between the stable and the volatile sub-session is in accordance with just such a shift in behavior. The exact mechanism by which the broad spectrum of LR parameters present in dACC, concerning many possible choice values estimated at different timescales, is integrated into one eventual decision needs further elucidation.

In conclusion, there are multiple experience dependent value estimates with coarse but systematic topographies in dACC and three other regions. Interactions between these regions occur in relation to this pattern of specific timescales. The distributions of value estimates are dynamically adjusted when there are changes in the environment's volatility. The dynamic increase in signal strength (or signal-to-noise ratio) in voxels representing behaviorally relevant LRs in the ACC might be due to more neurons reflecting the information (possibly due to reverberant local circuit activity) or due the same number of neurons reflecting the information but with less noise (i.e., change in firing rates), or a combination of both mechanisms. These changes would align with recurrent neural network accounts of information processing with mutual inhibition[43] that argue that such local networks can maintain information about previous states and serve as substrates for decision making. Dynamic adjustment based on environmental statistics might be critical for adjusting behavior to a particular LR and for selecting a particular choice on a given trial.

## Methods

**Behavioral task and fMRI.** The behavioral task and scanning procedures have been described in detail before[1]. In the task, subjects were presented with two choice options, a green and a blue rectangle (Fig. 2a). The potential reward magnitudes were presented in the centre of each stimulus while the reward probabilities had to be learned by the subjects. Reward probabilities were changing throughout the experiment. There was a stable sub-session of 60 trials where one of the stimuli was rewarded 75% of trials and the other one 25% and a volatile sub-session where reward probabilities for the stimuli were 80 and 20%, changing every 20 trials. The order of the sub-sessions was counterbalanced between subjects. Reward information was coupled between the stimuli, i.e., the feedback that the chosen stimulus was rewarded also implied that the choice of the other stimulus would not have led to a reward, and vice versa. If the chosen stimulus was rewarded, the presented reward magnitude was added to the subjects accumulating points and a red bar at the bottom of the screen increased in proportion to the points acquired. When the red bar reached a vertical silver bar, subjects received £10, if it reached a golden bar, they received £20 at the end of the experiment. Subjects were presented with the two options for 4–8 s (jittered). When a question-mark appeared, they could signal their choice with a button press. As soon as the button press was registered, subjects had to wait for 4–8 s (jittered) until the rewarded stimulus was presented in the middle. After a jittered inter-trial-interval of 3–7 s, the next trial began. EPI images were acquired at 3 mm³ voxel resolution with a repetition time (TR) of 3.0 s and an echo time (TE) of 30 ms, a flip angle of 87°. The slice angle was set to 15° and a local z-shim was applied around the orbitofrontal cortex in order to reduce signal drop-out[1]. Since the response was self-timed, the experiment's duration was variable. On average, 830 volumes (41.5 min) were acquired. A T1 structural image was acquired with an MPRAGE sequence with 1 mm³ voxel resolution, a TE of 4.53 ms, an inversion time(TI) of 900 ms and a TR of 2.2 s[1].

We used FMRIB's Software Library (FSL)[53] for image pre-processing and the first level data analysis (Supplementary Method 1). Subsequent analysis steps relating to the LR regressors were performed with MATLAB (R2015a 8.5.0.197613).

The preprocessing was performed on the functional images of the entire session (for the initial analysis), and of the stable and the volatile sub-sessions (for subsequent analyses). In order to analyze the sub-sessions, we split the time series of BOLD data into those portions that were collected when the reward environment was in a stable or volatile sub-session. The data assigned to the first sub-session encompassed all MRI volumes collected up to and including the onset of the last outcome of that sub-session of the experiment plus two additional volumes to account for the delay of the hemodynamic response function.

The data were pre-whitened before analysis to account for temporal autocorrelation[54]. For the subsequent mapping of LRs, we ran three GLM's for the whole session, and separately for the stable and the volatile sub-sessions, at the first level for each participant with the following regressors:

(1) Decision phase main effect (duration: stimuli onset until response)
(2) Predict phase main effect (duration: response until outcome)
(3) Outcome monitor phase main effect (duration: 3 s)
(4) Parametric modulation of decision phase with reward magnitude of chosen stimulus
(5) Parametric modulation of decision phase with log of reaction time
(6) Parametric modulation of decision phase with stay (0) or switch (1) decision
(7) Parametric modulation of outcome monitor phase with the reward magnitude of the chosen stimulus

We also added the temporal derivative of each regressor to the design matrix in order to explain variance related to possible differences in the timing between the assumed and the actual hemodynamic response function (HRF).

Since reward magnitudes are changing unpredictably, participants estimate reward probabilities and not action values. Thus, for each subject, we then calculated the probability estimates for each stimulus from a simple reinforcement learning model[55], based on all 99 LRs ($\alpha$) between 0.01 and 0.99. The model estimates the probability of one of the stimuli leading to a reward by updating the stimulus-reward probability $p(a)$ with LR $\alpha$, where $R = 1$ when the stimulus was rewarded and $R = 0$ if not:

$$p(a_i) = p(a_{i-1}) + \alpha[R - p(a_{i-1})].$$

The probability estimate of the other stimulus $p(B)$ is $1 - p(A)$. From these values, we also calculated the prediction error (PE) corresponding to the outcome of that trial by subtracting the probability estimate of the chosen stimulus from the outcome (1 for rewarded trials, 0 for non-rewarded trials). Thus, the PE is a "probability PE" that is not weighted with the magnitude of the (foregone) reward. After normalizing the probability estimates for all LRs for both stimuli, we derived the probability estimate of the chosen stimulus p(chosen). These p(chosen)-regressors (hereafter "LR regressors") and the PE regressors were convolved with the HRF, normalized and high-pass filtered in the same way (in the same manner as in FSL). We calculated a correlation matrix for the 99 resulting LR regressors for every subject and for the whole session as well as the two sub-sessions. Since the correlation between regressors is not the same for all levels of LR, we chose 30 regressors that were equally spaced in terms of their correlation to the neighboring regressors. We did so by averaging the 30 LR regressors with equal correlation for every subject in all three sessions and subsequently rounding them to two decimals. This procedure resulted in 30 LR regressors corresponding to the following LRs (see also Fig. 2):

[0.01 0.02 0.03 0.04 0.05 0.06 0.07 0.08 0.09 0.11 0.12 0.14 0.15 0.17 0.20 0.22 0.25 0.28 0.32 0.36 0.40 0.46 0.51 0.57 0.64 0.71 0.78 0.85 0.93 0.99].

We used the BET procedure[56] on the high-pass filtered and motion corrected functional MRI data to separate brain matter from non-brain matter. For each of the (sub-)sessions in every subject, we explained activity in the filtered fMRI data with 30 separate GLM's, each with the design matrix described above together with one of the 30 LR regressors (onset during the decision phase) and the corresponding PE regressor (onset during outcome monitoring phase).

In each GLM, we retrieved the parameter estimate for the LR regressor and we mapped the following three measures to every voxel in the brain:

(1)    Best-fitting LR: the regressor with the highest beta-value (regression weights indicative of the relationship between the regressor and the BOLD signal) in the GLM. For example, if regressor 20 had the highest beta-value amongst the 30 LR regressors, that voxel would be assigned a LR of 20.
(2)    The change in the best-fitting LR between the stable and the volatile sub-sessions (measured as best-fitting LR in the stable sub-session minus the best-fitting LR in the volatile sub-session)
(3)    The beta-weight of the best-fitting LR regressor in the entire session and in the stable and the volatile sub-sessions

The resulting images were registered to MNI-space using the nonlinear warping field using nearest-neighbor interpolation. Subsequently, the single-subject images were averaged across all subjects to create group-average images.

We also used a standard FSL analysis with a GLM similar to the one above but with two additional regressors corresponding to the probability of the chosen stimulus during the decision phase and during the outcome monitoring phase as derived from a Bayesian learner model[1], as well as a regressor coding the outcome of the trial (won or lost). This analysis was used for generating regions of interest (ROIs; Fig. 3) with the regressor of the magnitude of the chosen option's potential reward. Note that this is the only GLM where the Bayesian learning model was used, and it was only used in order to include value estimates while not having to arbitrarily choose one of the 30 LR-regressors from the reinforcement learning model.

We defined our ROIs by the overlap of the contrast over this regressor (cluster-corrected results with the standard threshold of $z = 2.3$, corrected significance level $p = 0.05$) and anatomical masks derived from the connectivity-based parcellation atlases[19,20] (http://www.rbmars.dds.nl/CBPatlases.htm) (Fig. 3). For dACC, this included bilateral areas 24a/b, d32 as well as the bilateral anterior rostral zones of the cingulate motor areas. For posterior vmPFC, this included bilateral area 14 m and for anterior vmPFC it included 11 m[20]. For IPL, this included inferior parietal lobule areas c and d as defined by Mars et al.[19]. The atlas only contains IPL regions for the right hemisphere, we therefore mirrored the regions along the midline to create masks for the left hemisphere. Since the anatomical masks are defined by white matter connectivity, they do not cover the entire cortical area. Therefore, the dACC and vmPFC masks were extended with 2 voxels medially, while the IPL masks were extended laterally and caudally to ensure that all gray matter voxels were covered by the masks.

**Voxel activity reflecting reinforcement learning**. In order to confirm that the voxels in our ROIs actually reflected activity that was related to probability

estimates, we ran a singular value decomposition (SVD) over the 99 LR regressors (before HRF-convolution, normalization and high-pass filtering) to derive singular values capturing most of the variance associated with the variability in the 99 LR regressors. For every voxel, we then derived the Akaike Information Criterion (AIC) scores from our main GLM (not containing any LR regressors) as well as from a GLM that contained the first three singular values from the SVD (HRF-convolved, demeaned and high-pass filtered). We then used random-effects Bayesian model comparison for group studies[21] by passing each subject's AIC scores for the two models to the spm_bms matlab function from SPM12 (http://www.fil.ion.ucl.ac.uk/spm/software/spm12/). This procedure returned protected exceedance probabilities for every voxel, showing the probability that the model containing the singular values was a more likely model of the data than the model without those components.

**Data availability**. The data that support the findings from this study are available from the corresponding author upon request.

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

## Acknowledgements

Work was funded by the Wellcome Trust (M.F.S.R.: WT100973AIA; M.K.W.: 096589/Z/11/Z) and the NovoNordiskFoundation (NNF14OC0011413). N.K. is a Christ Church Junior Research Fellow. L.V. held a Marie Curie fellowship.

## Author contributions

D.M. and L.V. analyzed data; T.E.J.B. acquired the data; D.M., K.H.M., O.J.H., N.K., L.V., M.K.W., J.S. and M.F.S.R. developed the analysis approach; D.M., N.K., L.V., M.K.W., J.S., K.H.M, O.J.H., T.E.J.B. and M.F.S.R. discussed the results and wrote the manuscript.

## Additional information

**Competing interests:** The authors declare no competing financial interests.

