## [Peer Review File · Nature Communications]

Reviewers' comments:

Reviewer #1 (Remarks to the Author):

Meder et al tested subjects with the same behavioral task previously used by Behrens et al 2007. Subjects had to choose between two options identified by two visual stimuli and rewarded with high/low probability (75%/25% or 80%/20%). Sessions were divided in two sub-sessions – a stable one in which the stimulus-probability association was stable, and a volatile one in which the stimulus-probability association was reversed every 20 trials. fMRI scans were obtained during task performance. The novelty of the study is largely in the analysis approach. The authors focused on predefined ROIs in the dorsal anterior cingulate cortex (dACC) and in the parietal lobule (IPL), assuming that these are the hot spots of learning in this task. For each voxel, they ran a glm that included terms for the estimated reward probability and the reward prediction error. The latter term depends on a learning rate (LR), which in principle can vary between 0 and 1. Independent of the behavior, the authors ran 30 glm models for LRs that spanned the interval .01-.99, and they identified the best-fitting LR. They then defined a parameter termed "exceedance probability". The rationale and procedures used to define this parameter are presented very poorly in the ms. However, out of it comes, for each voxel, an assessment of whether LRs "have an impact" on the BOLD signal in that voxel. The next step was to examine, for each ROI, how best-fitting LRs were distributed topographically in cortex. Again, the authors do a poor job explaining the procedures used for this analysis. For what it is worth, they conclude that there is a topographic organization in dACC but not in IPL. This organization is the main conclusion of the study.

From my perspective, it is very hard to make sense of this paper because the details are presented in a cryptic, trust-me-and-don't-worry-about-it way. For the part that I do understand, there seem to also be other problems. For example the results shown for individual subjects suggest that the topographic organization in dACC might be an artifact of averaging across subjects.

Major concerns

1. I consider myself fairly literate in linear statistics and computational methods more generally. Still, I cannot make sense of what the authors write about the exceedance probability (lines 188-209). Specifically, I don't understand either the rationale or the procedures used to define this particular measure. Because this parameter is so central to the analysis, it is simply impossible for me to evaluate the results of this study.

2. I also do not understand the procedures leading to the conclusion that there is a topographic organization in dACC (lines 244-256). This description, like that of the exceedance probability, is totally cryptic.

3. From the little I do understand, I have the impression that the procedure used for exceedance probability has some degree of double dipping. That is because the authors ran for each voxel 30 LR models. Therefore, the best-fitting LR model almost certainly over-fits the subject data. As a result, the analysis based on the exceedance probability seems biased towards the statement that "LR have an impact on activity". One way to address this issue would be to use different trials to fit the LR models and to analyze the exceedance probability.

4. One puzzling aspect of the study is the a-priori focus on dACC and IPL. In a value learning / value-based decision task like this, there are many other candidate areas that seems worth considering, including ventral tegmental area, striatum, lateral prefrontal cortex, ventromedial prefrontal cortex, orbitofrontal cortex, amygdala, cerebellum -- and other. Why were ROI defined in such a restrictive way? And if whole-brain analyses were ran (which the supplement somewhat suggests), then how did

the procedure differ between the analyses ran for ROIs and the analyses ran for the rest of the brain?

5. Fig.4b. If these are the results for the two best subjects, the conclusion is not convincing. For example, the second subject seems to show a bimodal distribution rather than a gradient. More generally, the topographic organization in dACC might be an artifact of averaging across subjects. The authors should show and analyze the results for every subject included in the study.

6. Even if one gives the results the benefit of the doubt, I don't see any connection between the topographic distribution of best-fitting LRs and any behavioral measure of learning rate. Beyond other issues, this calls into question the significance of this study.

Reviewer #2 (Remarks to the Author):

In their manuscript, Meder et al. reanalyze fMRI data from a probabilistic learning/decision-making task to determine whether the brain represents information about the history of received rewards over multiple time scales. In the task, participants made a series of two alternative choices in which reward magnitudes changed randomly and were displayed on the screen, but hidden reward probabilities drifted and could be tracked by updating internal estimates of the probability associated with each stimulus. The authors report that individual voxels in dorsal anterior cingulate cortex (dACC) and inferior parietal lobule (IPL) ROIs reflect choice variables derived from different learning rate (LR) parameters, and within an ROI there is subregional variation in the temporal integration of past rewards. Multiple time constants in neural activity have been reported previously, but the authors extend these findings in a few important ways. For instance, the time constant appeared to vary systematically with anatomical positions within the ROIs, and the activity of voxels with similar time constants tended to covary, suggesting functional interactions. Overall, I think the results are intriguing, but need more investigation to make sure they hold up under scrutiny.

Major comments

(1) In describing how they arrived at optimal LR parameters, the authors state the following: "The best-fitting LR of a voxel corresponds to the value regressor calculated with an LR that explained most of the variance in the voxel's time-course, compared to other LR regressors, regardless of how much variance it actually explains." Did the variance explained differ across voxels in any systematic way? In other words, it seems clear that for some voxels the optimal LR was large, meaning they depended more on recently experienced outcomes, and these tended to be anatomically clustered. But could it be that the small LR voxels, the ones that depend less on recently experienced outcomes, are just poorly fit by learning-related variables in general, such that flatter functions are found to be "optimal", while not really explaining much variance in the signal? This is important because, if this alternative were to be the case, what you have is not multiple LRs, but subregional localization of learning-related changes.

(2) Related to the comment above, how reliable are these optimal LRs in the data? Ideally one would use cross-validation within subject to see whether different folds of data give rise to similar LR estimates.

(3) In Figure 8 and the corresponding supplemental figure, there are correlations for high LR voxels but this doesn't hold up consistently at lower LRs. Has this trend been investigated? Could it be related to the suggestion in (1), that the low LR voxels may not be describing real variability in the data, and therefore don't relate to other brain regions with similar low LRs?

Minor comments

Figure 1 – the legend doesn't really describe what is shown in the figure (e.g. what does "value retrieval: valueA – valueB" mean, and how does it fit into what is being conveyed in the figure?). I'm not sure what this figure contributes beyond what the reader can understand from the text alone.

Figure 2A – the legend should provide a more thorough description of the task features being illustrated. E.g. I gather that the gray outline indicates which option was chosen?

I'm a bit confused by the "signal strength hypothesis". It is described as the case where the value estimates remain constant but the signal strength of some voxels increase. What do the authors hypothesize is changing physiologically? More neurons with a particular selectivity are recruited? Neurons change their firing rates to a greater degree?

In the figures, it would be helpful to label the color bars/axes according to the actual LRs (0 to 1), since that has a direct interpretation in the data, rather than the identifying number (1 to 30), which is less intuitive.

Reviewer #3 (Remarks to the Author):

Meder and colleagues report results showing topographical encoding of option values with different learning rates (LR) in the dorsal anterior cingulate cortex (dACC) and inferior parietal lobule (IPL) in humans performing a probability shifting task with different levels of volatility as in Behrens and colleagues 2007. The authors performed a computational analysis of brain activity independently of model fitting to the behavior. Such a modeling approach allows for the search of neural correlates of different model parameters (here LR) which are not necessarily reflected in the subjects' behavior.

The authors found evidence for multiple values with different time constants (i.e. putatively learned with different LRs) within the decision-aligned activity of dACC and IPL. They moreover found that the specific LR encoded by a particular voxel within these regions would increase under high volatility and decrease under low volatility. In addition, they report an increase specific to dACC in the correlation between the best-fitting LRs and their beta-weights under high volatility compared to low volatility. In contrast, vmPFC showed no shift in the distribution of LRs depending on the volatility of the environment. Finally, focusing on the residuals of the previous correlations, they found that voxels in regions such as dACC, IPL and frontopolar cortex preferentially interacted with voxels with similar LRs in other brain regions within this set. In contrast, vmPFC or VS did not show such a selectivity, illustrating that this is not a general tendency of interaction between brain regions nor due to noise but a specific effect to multiple value encoding regions.

The approach and results are very interesting but some aspects of the methods may not be appropriately controlled.

Major Points

1. I found slightly unfair the computation of exceedance probability of a GLM with regressors from all 30 LRs plus singular values from SVD compared with a GLM with the absence of any LR regressors. In the latter GLM, why not also performing SVD based on the basic regressors and including the singular values? Could the authors also give the exceedance probability of this comparison when the first GLM does not contain SVD singular values? In addition to comparing multiple LRs with no LRs, I think the authors should also show evidence for multiple LRs against a single best LR. Why not then also testing alternative GLMs such as one containing a single LR which best fits the behavior of the subjects? And

another GLM using a single LR explaining the highest variance of dACC activity? Could the authors show exceedance probability for each comparison of pairs of GLMs but also for the global comparison of all above mentioned GLMs? Finally, did the author include the prior over model parameters in the computation of the exceedance probability (which should be computed based on Bayesian evidence rather than likelihood)?

2. The authors claim using a 'new approach' to analyse neural data with computational models independently of behavior fitting. However, other groups have already performed similar approaches which should be cited here: Lee, S. W. W., Shimojo, S. & O'Doherty, J. P. Neural Computations Underlying Arbitration between Model-Based and Model-free Learning. *Neuron* 81, 687–699 (2014). Wilson, R. C. & Niv, Y. Is model fitting necessary for model-based fMRI?. *PLoS Comput Biol* 11.6: e1004237 (2015). I think that other groups have subsequently used this method such as the group of Bernard Balleine.

3. Did the authors use corrections for multiple comparisons when searching for correlations between voxels and the 30 different LRs? Especially when looking for a gradient within dACC?

4. The current formulation is sometimes unclear as to whether the authors used as regressors the probability of the chosen option and the prediction error (PE) from a classical reinforcement learning model or from the Bayesian learner model of Behrens and colleagues (2007). The ambiguity lies on the fact that the authors seem to alternate between analyses with the two models. Although the methods are sometimes clearer (stating that the probability of chosen option and PE used in the design matrix with 30 different LRs come from a reinforcement learning model), I think it would help the reader to make it clearer throughout the manuscript. For instance, when presenting the results in Figure 2. Also when stating that the authors did not replicate the results of Behrens and colleagues (2007) about PE encoding in the dACC. Could this be due to the authors using a reinforcement learning model rather than the Bayesian learner? Since the authors use both models here, could they please make sure they replicate the results of Behrens and colleagues (2007) with the Bayesian learner, and if not, could they discuss and interpret it?

5. How circular is the calibration of the gap between LRs as a function of the correlation between these regressors with respect to each subject's brain activity? Could the authors instead a priori calibrate the gap between LRs in order to equalize the correlation between the model timeseries generated by model simulation independently of subjects' behavior and brain activity?

6. Figure 8B: could the authors interpret why the matrix contains high correlation values in the borders as well as in horizontal line #18 (dACC) and vertical lines #7 and #19 (IPL)?

7. Line 426, the authors should show the stats (incl. exceedance probability) when referring to the results of the analysis for the vmPFC.

8. I found very interesting and important the analysis showing interaction between voxels of different regions with similar LRs in contrast to regions which do not show such an effect. Since such an effect may reflect parallel estimations which at some point need to converge/integrate in order to make a decision, I am wondering whether the authors could push further the analysis to identify potential regions where the analysis would suggest a convergence/integration. This would definitely strengthen the whole demonstration.

9. Also in terms of interpretation, could the authors discuss their results with respect with other types of gradients reported in terms of reinforcement learning models? Tanaka, S. C., Doya, K., Okada, G., Ueda, K., Okamoto, Y., & Yamawaki, S. Prediction of immediate and future rewards differentially

recruits cortico-basal ganglia loops. *Nature neuroscience*, 7(8), 887-893 (2004). Holroyd, C. B., & McClure, S. M. Hierarchical control over effortful behavior by rodent medial frontal cortex: A computational model. *Psychological review*, 122(1), 54 (2015).

Minor Points

1. Since there is no value update hypothesized by reinforcement learning models at the decision time, to make the sentence in line 52 clearer, the authors could reformulate in the following manner: 'It is therefore unclear how dACC activity at the point in time when decisions are actually made change as a function of etc.'
2. Along the same lines, line 69: 'calculate a single value' -> 'consider a single previously calculated value'.
3. What are the links and differences between Figure 4A and Figure 2E? Do they come from the same analysis? This is not clear to the reader.
4. Figure 7B: why not trying a linear regression and if it is significant, showing that the slope is negative in the upper panel and positive in the lower one?
5. Line 696: enssure -> ensure.
6. Supplementary material lines 158 and 160: Fig. S8A -> Fig. S6A and Fig. S8B -> Fig. S6B.

Rebuttal

We sincerely appreciate Reviewer 2's and 3's careful and constructive evaluation, which helped us to considerably improve our manuscript. We have introduced several new analyses and clarifications following their suggestions. However, we note that criticisms made by reviewer 1 are factually incorrect. These inaccuracies and the discrepancy between reviewer 1's review and the other reviews have led us to have concerns about the fairness of the reviewer 1's review. We have included a detailed item-by-item response below in which the reviewers' comments are shown in black ink, our replies in blue italics, and the text from the revised manuscript in blue ink in a standard font.

Reviewer #1 (Remarks to the Author):

Meder et al tested subjects with the same behavioral task previously used by Behrens et al 2007. Subjects had to choose between two options identified by two visual stimuli and rewarded with high/low probability (75%/25% or 80%/20%). Sessions were divided in two sub-sessions – a stable one in which the stimulus-probability association was stable, and a volatile one in which the stimulus-probability association was reversed every 20 trials. fMRI scans were obtained during task performance. The novelty of the study is largely in the analysis approach. The authors focused on predefined ROIs in the dorsal anterior cingulate cortex (dACC) and in the parietal lobule (IPL), assuming that these are the hot spots of learning in this task. For each voxel, they ran a glm that included terms for the estimated reward probability and the reward prediction error. The latter term depends on a learning rate (LR), which in principle can vary between 0 and 1. Independent of the behavior, the authors ran 30 glm models for LRs that spanned the interval .01-.99, and they identified the best-fitting LR. They then defined a parameter termed "exceedance probability". The rationale and procedures used to define this parameter are presented very poorly in the ms. However, out of it comes, for each voxel, an assessment of whether LRs "have an impact" on the BOLD signal in that voxel. The next step was to examine, for each ROI, how best-fitting LRs were distributed topographically in cortex. Again, the authors do a poor job explaining the procedures used for this analysis. For what it is worth, they conclude that there is a topographic organization in dACC but not in IPL. This organization is the main conclusion of the study. From my perspective, it is very hard to make sense of this paper because the details are presented in a cryptic, trust-me-and-don't-worry-about-it way. For the part that I do understand, there seem to also be other problems. For example the results shown for individual subjects suggest that the topographic organization in dACC might be an artifact of averaging across subjects.

We are concerned that many of the comments made by reviewer 1 are factually inaccurate and at odds with those made by the other authors. We have nevertheless acted as if the points were made in good faith and have attempted to make sure that there is no room for any misunderstanding in the revised manuscript. In order to respond most effectively we have responded to the reviewer's major points in a slightly different order to the one in which they were originally made.

Major concerns

4. One puzzling aspect of the study is the a-priori focus on dACC and IPL. In a value learning /

value-based decision task like this, there are many other candidate areas that seems worth considering, including ventral tegmental area, striatum, lateral prefrontal cortex, ventromedial prefrontal cortex, orbitofrontal cortex, amygdala, cerebellum -- and other. Why were ROI defined in such a restrictive way? And if whole-brain analyses were ran (which the supplement somewhat suggests), then how did the procedure differ between the analyses ran for ROIs and the analyses ran for the rest of the brain?

Reviewer 1's statements about our manuscript are factually inaccurate. Contrary to reviewer's claim we presented extensive analyses of three areas the reviewer mentions: anterior lateral prefrontal cortex, ventromedial prefrontal cortex (vmPFC,) and ventral striatum. A detailed discussion of these results occupied more than three pages of text on pages 20-22 in the main manuscript (now pages 23-25 in the revised manuscript) and most of the seven pages of Supplementary Information. In fact these sections are so extensive that it is difficult to include them here in the letter of rebuttal. Moreover the results were illustrated in a series of six figures (S1-S6). We note that these analyses were referred to by the other reviewers in their review. The level of inaccuracy in reviewer 1's review raises concerns about its fairness.

We agree with the reviewer that other brain areas beyond ACC are involved in aspects of decision making but we disagree with the reviewer in thinking that such areas are all equally good candidates for carrying the type of information we discuss. Nevertheless we have tried in the revised manuscript to more clearly explain our reasons for focusing on the ACC and IPL as follows:

First, we considered the dACC because of its role both in monitoring feedback and decision outcomes and in adaptive control of subsequent behaviour and because its outcome-related activity is known to be related to the learning rate dACC^{1,8-15}. It is also the area in which activity has been most consistently related to adjustment in the speed of behavioral change. In addition we considered the inferior parietal lobule (IPL) because it is frequently co-activated with the dACC during decision making tasks¹⁶.

Secondly, while it is true that we proposed regions of interest (ROIs) in the Introduction this was not the only approach that we took to identifying the location of learning rate (LR)-related activity. We also used a whole brain analysis to identify brain areas carrying LR-related information. We did not, however, do this by mapping best-fitting LRs across the whole brain as the reviewer reports. Instead we ran a whole-brain analysis for a model comparison in which we compared models that described brain activity either with and without a set of singular values summarizing the LRs. The nature of the analytical approach we took, as opposed to the one that the reviewer said we took, is important because it avoids the "double dipping" and "over-fitting" concerns that would be associated with the other approach. The result of the analysis was important in order both to ascertain that our a-priori ROIs specifically showed high evidence for LR-related information processing and that this was not the case for all brain regions such as the vmPFC and ventral striatum. The model comparison also revealed two further regions (rFO and FPl) with high model evidence for the model with singular values. Since we did not have these regions as a-priori defined ROIs, we did not perform any of the analyses run on ACC and IPL, except for the time-course correlations. We deemed this important in order to corroborate that our finding that LRs are an organizing principle of inter-regional connectivity was a replicable result.

1. I consider myself fairly literate in linear statistics and computational methods more generally. Still, I cannot make sense of what the authors write about the exceedance probability (lines 188-209). Specifically, I don't understand either the rationale or the procedures used to define this particular measure. Because this parameter is so central to the analysis, it is simply impossible for me to evaluate the results of this study.

We note that the approach that we have used, which is based on exceedance probabilities, is an approach that is used commonly in Bayesian statistics and we note that it was clearly understood by the other reviewers. Nevertheless we have tried to improve the explanation of the concept of exceedance probabilities as well as the rationale behind the choice for this analysis approach (additional information in bold) below:

We also used a second independent approach to identify ROIs. This approach identified very similar areas in dACC and IPL (Fig. 3B) and in two other brain regions (Fig. S1). The analysis approach that we have described so far (Fig. 2) assigns a best-fitting LR in comparison to all other LR regressors, but it does not quantify whether this best-fitting regressor explains any significant amount of evidence. **In order to confirm that the voxels in our ROIs reflected activity that was related to probability estimates in a manner that could not be due to the overfitting of any particular LR in a given voxel**, we ran a singular value decomposition (SVD) over the LR regressors (before HRF-convolution, normalisation and high-pass filtering) to derive singular values capturing most of the variance associated with the LR regressors. **The first three singular values explained on average 99.63% of variance in the LR regressors and can thus be interpreted as capturing almost all LR-related variance.** A standard t-test over the parameters of these three regressors in a GLM explaining the measured BOLD signal in each voxel is not an appropriate test because it assumes a coherent positive or negative correlation between the regressor and the measured BOLD signal in a given voxel across all subjects. **An F-test over the parameters might therefore be appropriate but there is currently no widely agreed standard approach for combining individual subject F-test results into a group-level analysis in neuroimaging.** We therefore used a different approach to test whether a voxel's time course reflected LR-related information. For every voxel, we derived the Akaike Information Criterion (AIC) scores from our main GLM **(the seven regressors of no interest and their temporal derivatives (see Methods) plus six motion confound regressors, but in the absence of any LR regressors)**. This reveals how well a model lacking multiple LRs accounts for activity variation in every voxel in the brain. We also ran an identical GLM that contained the same regressors but also the first three principle components from the SVD (HRF-convolved, demeaned and high-pass filtered), and again computed the AIC score. This reveals how well a model containing LR-based reward probability estimates accounts for activity variation in every voxel in the brain. We then compared the AIC scores of the two models of brain activity at every voxel using random-effects Bayesian model comparison for group studies²¹. This procedure returned protected exceedance probabilities for every voxel, revealing the degree to which the model containing the singular values, reflecting value estimates based on one or multiple LRs, was the more likely model of the neural data (Fig. 3B). **The protected exceedance probability can be considered a quantitative measure of the evidence for the model with LR-related information. The random effects**

Bayesian model comparison tests for the amount of evidence in favour of a model across all subjects, regardless of sign. Since exceedance probabilities of all compared models sum to 1, they can be easily interpreted in a way that is similar to classical p-values^{21,22}. In voxels with a protected exceedance probability of >95%, this corresponds to a 95% confidence that the model with singular value regressors is the most likely model to explain the activity. Thus, we can state that in these voxels LRs have an impact on activity.

2. I also do not understand the procedures leading to the conclusion that there is a topographic organization in dACC (lines 244-256). This description, like that of the exceedance probability, is totally cryptic.

We note again that the approach that we have used here is a common regression analysis and we note that it was clearly understood by the other reviewers. However, once again, we have tried to elaborate on the explanation of the approach (additional information in bold) so that it will be clear to the widest group of readers:

Using our multivariate mapping approach, we found that in our ROIs, voxels did not homogeneously integrate the reward history with the same LR, but that there was some degree of spatial topographic organization of the diverse probability estimates (Fig. 4). **First, we asked whether the spatial distribution of the best-fitting LRs was entirely random or whether it was organized in three-dimensional space, we tested whether the voxels' x, y, and z-coordinates (Montreal Neurological Institute (MNI) space) could predict the best-fitting LR. While these axes do not correspond exactly to anatomical features such as gyri or sulci in these regions, this analysis is a first indication of spatial organization.** In both IPL and dACC, a significant amount of variability in the best-fitting LRs in voxels was explained by the x, y, and z coordinates of the voxel when regression models were fitted to each subject's data (t-test over the variance explained by every subject's regression model (r^2) against the mean r^2 of 10,000 regression models with randomly permuted coordinates. dACC: Mean r^2 true data = 0.101, mean r^2 permuted data = 0.002, $t_{16} = 5.071$, $p < 0.001$, IPL right hemisphere: Mean r^2 true data = 0.124, mean r^2 permuted data = 0.003, $t_{16} = 5.566$, $p < 0.001$, IPL left hemisphere: Mean r^2 true data = 0.182, mean r^2 permuted data = 0.006, $t_{16} = 5.040$, $p < 0.001$). The principle axis of anatomical organization in dACC in humans and other primates is approximately rostrocaudally oriented^{20,30}. Although this axis does not fully correspond to the cardinal axes in MNI space we nevertheless examined whether LRs were also organized along the MNI y-axis. Consistently, across subjects, in the dACC, LRs showed a gradient along the MNI y-axis with increasing LRs in the rostral direction (t-test of subjects' regression coefficients of the y-coordinate regressor against 0, $t_{16} = 2.175$, $p = 0.045$). No major direction of anatomical organization has been reported for the IPL, **thus here we did not test for a spatial gradient along any cardinal axis from MNI space.**

3. From the little I do understand, I have the impression that the procedure used for exceedance probability has some degree of double dipping. That is because the authors ran for each voxel 30 LR models. Therefore, the best-fitting LR model almost certainly over-fits the subject data. As a result, the analysis based on the exceedance probability seems biased towards the statement that "LR have an impact on activity". One way to address this issue would be to use different trials to fit the LR models and to analyze the exceedance probability.

The analysis approach does precisely the opposite to what the reviewer suggests. Maybe this confusion stems from the fact that the reviewer did not report our analysis procedure correctly (see above response to point 4). The approach is designed to avoid the risk of double-dipping. The fact that fitting 30 different models to the activity in a voxel might lead to overfitting is exactly the reason why we used the random-effects Bayesian model comparison approach with singular value regressors. As explained above, the first three singular values summarize the entire set of LRs. This allowed us to confirm that the voxels in our ROIs reflected activity that was related to probability estimates without overfitting the data. The first three singular values from the SVD reflect common variance related to probability estimates based on reinforcement learning, unspecific to one single LR. We have tried to make this clearer in the revised description of our approach so that there is no room for ambiguity.

We also used a second independent approach to identify ROIs. This approach identified very similar areas in dACC and IPL (Fig. 3B) and in two other brain regions (Fig. S1). The analysis approach that we have described so far (Fig. 2) assigns a best-fitting LR in comparison to all other LR regressors, but it does not quantify whether this best-fitting regressor explains any significant amount of evidence. **In order to confirm that the voxels in our ROIs reflected activity that was related to probability estimates in a manner that could not be due to the overfitting of any particular LR in a given voxel**, we ran a singular value decomposition (SVD) over the LR regressors (before HRF-convolution, normalisation and high-pass filtering) to derive singular values capturing most of the variance associated with the LR regressors. **The first three singular values explained on average 99.63% of variance in the LR regressors and can thus be interpreted as capturing almost all LR-related variance. A standard t-test over the parameters of these three regressors in a GLM explaining the measured BOLD signal in each voxel is not an appropriate test because it assumes a coherent positive or negative correlation between the regressor and the measured BOLD signal in a given voxel across all subjects. An F-test over the parameters might therefore be appropriate but there is currently no widely agreed standard approach for combining individual subject F-test results into a group-level analysis in neuroimaging.** We therefore used a different approach to test whether a voxel's time course reflected LR-related information. For every voxel, we derived the Akaike Information Criterion (AIC) scores from our main GLM (**the seven regressors of no interest and their temporal derivatives (see Methods) plus six motion confound regressors, but in the absence of any LR regressors**). This reveals how well a model lacking multiple LRs accounts for activity variation in every voxel in the brain. We also ran an identical GLM that contained the same regressors but also the first three principle components from the SVD (HRF-convolved, demeaned and high-pass filtered), and again computed the AIC score. This reveals how well a model containing LR-based reward probability estimates accounts for activity variation in every voxel in the brain. We then compared the AIC scores of the two models of brain activity at every voxel using random-effects Bayesian model comparison for group studies²¹. This procedure returned protected exceedance probabilities for every voxel, revealing the degree to which the model containing the singular values, reflecting value estimates based on one or multiple LRs, was the more likely model of the neural data (Fig. 3B). **The protected exceedance probability can be considered a quantitative measure of the evidence for the model with LR-related information. The random effects Bayesian model comparison tests for the amount of evidence in favour of a model across all subjects, regardless of sign. Since exceedance probabilities of all compared models sum to 1, they can be easily interpreted in a way that is similar to classical p-values^{21,22}. In voxels with a protected exceedance probability of >95%, this corresponds to a 95% confidence that the model with singular value regressors is the most likely model to explain the activity.** Thus, we

can state that in these voxels LRs have an impact on activity.

5. Fig.4b. If these are the results for the two best subjects, the conclusion is not convincing. For example, the second subject seems to show a bimodal distribution rather than a gradient. More generally, the topographic organization in dACC might be an artifact of averaging across subjects. The authors should show and analyze the results for every subject included in the study.

The first draft of the manuscript made clear that all subjects were already included in our analyses. Equally, it made clear that the topographic organization in dACC is not a result of averaging across subjects – as we stated in the manuscript, the finding is based on a “t-test of subjects’ regression coefficients of the y-coordinate regressor against 0, $t_{16} = 2.175$, $p = 0.045$ ”, thus it is a random-effects analysis conducted across all subjects that is significant despite some inter-individual variability. The two subjects shown are example subjects, shown in order to intuitively convey the gradient we have tested for (see below). We also remind the reviewer that where data points are shown in warmer yellow colours that this indicates that many data points were found at this position. Colder blue colours indicate lower numbers of data points at a given position. Moreover we note that we only claim “some degree of the diverse probability estimates” for IPL and ACC is organized topographically. All individual subject data summaries are now included in the Supplementary Information in the revised manuscript.

6. Even if one gives the results the benefit of the doubt, I don't see any connection between the topographic distribution of best-fitting LRs and any behavioral measure of learning rate. Beyond other issues, this calls into question the significance of this study.

This is incorrect. A major part of the manuscript on pages 13-18 and illustrated in figures 5, 6, and 7 examined changes in best fitting LRs in volatile and stable environmental situations that produce changes in learning rate (Behrens et al., 2007; 2008).

Reviewer #2 (Remarks to the Author):

In their manuscript, Meder et al. reanalyze fMRI data from a probabilistic learning/decision-making task to determine whether the brain represents information about the history of received rewards over multiple time scales. In the task, participants made a series of two alternative choices in which reward magnitudes changed randomly and were displayed on the screen, but hidden reward probabilities drifted and could be tracked by updating internal estimates of the probability associated with each stimulus. The authors report that individual voxels in dorsal anterior cingulate cortex (dACC) and inferior parietal lobule (IPL) ROIs reflect choice variables derived from different learning rate (LR) parameters, and within an ROI there is subregional variation in the temporal integration of past rewards. Multiple time constants in neural activity have been reported previously, but the authors extend these findings in a few important ways. For instance, the time constant appeared to vary systematically with anatomical positions within the ROIs, and the activity of voxels with similar time constants tended to covary, suggesting functional interactions. Overall, I think the results are intriguing, but need more investigation to make sure they hold up under scrutiny.

We thank the reviewer for their interest in the manuscript and have attempted to set out the results of the additional analyses that have been requested below.

Major comments

(1) In describing how they arrived at optimal LR parameters, the authors state the following: “The best-fitting LR of a voxel corresponds to the value regressor calculated with an LR that explained most of the variance in the voxel’s time-course, compared to other LR regressors, regardless of how much variance it actually explains.” Did the variance explained differ across voxels in any systematic way? In other words, it seems clear that for some voxels the optimal LR was large, meaning they depended more on recently experienced outcomes, and these tended to be anatomically clustered. But could it be that the small LR voxels, the ones that depend less on recently experienced outcomes, are just poorly fit by learning-related variables in general, such that flatter functions are found to be “optimal”, while not really explaining much variance in the signal? This is important because, if this alternative were to be the case, what you have is not multiple LRs, but subregional localization of learning-related changes.

We agree with the reviewer that these concerns are important, however, we believe that they can be addressed with some additional simple analyses and by explaining more clearly the implications of some of the existing analyses.

First, if there were a systematic variation in the variance explained by the LR such that higher LR explained more variance then there should be a correlation between the best-fitting LR for a voxel and the regressor’s beta weight at that voxel. There was no evidence for such a correlation when we analysed the entire data set (ACC: $r = 0.087$, $p = 0.150$; IPL; $r = 0.014$, $p = 0.812$).

Next we can look at the relationship between voxels’ LRs and the beta regression weights in more detail by breaking the data set up into two parts associated with the stable and variable task periods. In the variable condition, where faster LRs are more appropriate, there is a positive correlation between LR and regressor beta weight (mean z-standardized correlation values =

0.173) but by contrast in the stable condition, where slower LRs are more appropriate, the relationship is flipped (mean z -standardized correlation values = -0.058). This is illustrated in the figure below. In other words sometimes there is the relationship between LR and beta weight that the reviewer supposed but in an equal number of occasions the relationship reverses. The systematic manner in which the relationship reverses suggests that low LR do not uniformly explain less variance in a subset of voxels but rather that high and low LRs explain more or less variance in voxels depending on the task conditions. High LR voxels explain more variance when the subjects' learning rates are faster. Low LR voxels explain more variance when the subjects learning rates are slower.

Finally, we note a third reason why voxels that have a low learning rate should not be dismissed as failing to explain variance in the data. We note that the voxels in dACC assigned low learning rates are voxels where we found that it was necessary to include learning rate related information in order to explain variance in the BOLD signal; this is an implication of the model comparison illustrated in fig. 3b. Figure 3b shows that voxels assigned low learning rates are still voxels in which data is explained significantly better by a model with learning rate information than with no learning rate information.

(2) Related to the comment above, how reliable are these optimal LRs in the data? Ideally one would use cross-validation within subject to see whether different folds of data give rise to similar LR estimates.

We agree with the reviewer that some sort of cross-validation testing would have been interesting to perform. However, we have not found a way to cross-validate these time-series data since removing parts of the data for subsequent cross validation changes the entire structure of the time-series (and thus the best-fitting learning rate).

Instead we have attempted to address the reviewer's concerns in other's ways. The model comparison shown in figure 3 shows that a model of the data that includes LRs provides a better explanation of brain activity in dACC and IPL than a model without LRs. We have now added an additional set of model comparisons suggested by Reviewer 3 in which we compare models in which multiple learning rate models are compared against models employing the optimal behavioural LR. Again the evidence suggests a multiple LR model provides a better explanation of the fMRI data. We note that the LRs do not appear simply to reflect noise in the data because the LRs that are assigned change in a systematic way across the cortex (figure 4) and they change as task conditions change between stable and variable (figures 5, 6, and 7) and they are linked in a systematic way to the LRs encoded by other voxels in other brain areas (figure 8).

(3) In Figure 8 and the corresponding supplemental figure, there are correlations for high LR voxels but this doesn't hold up consistently at lower LRs. Has this trend been investigated? Could it be related to the suggestion in (1), that the low LR voxels may not be describing real variability in the data, and therefore don't relate to other brain regions with similar low LRs?

This point made by the reviewer is contingent on point 1 made by the reviewer that we have responded to in detail. As we argue above, in general, there is no evidence that less variance in BOLD is explained by low learning rates in voxels that are assigned low learning rates.

Minor comments

Figure 1 – the legend doesn't really describe what is shown in the figure (e.g what does “value retrieval: valueA – valueB” mean, and how does it fit into what is being conveyed in the figure?). I'm not sure what this figure contributes beyond what the reader can understand from the text alone.

We now explain the concepts shown in the figure legend more clearly:

When outcomes of decisions are witnessed, the prediction for the next choice is updated based on a learning rule where the prediction error (PE) is weighted by the learning rate a . Behrens et al. have shown that average activity in dACC reflects the environment's volatility and that under high volatility, the options' values are updated with a high learning rate a . However, at the time of the actual decision on the next trial, volatility no longer exerts a significant effect on average dACC activity. However, the representation of choice value estimates necessary for decision-making (the value estimate for option A relative to that of option B) might be represented in some other way such as an anatomically distributed pattern of activity where different value estimates might be calculated with different parameterizations of a , depending on the volatility.

Figure 2A – the legend should provide a more thorough description of the task features being illustrated. E.g. I gather that the gray outline indicates which option was chosen?

We have added a further sentence (bold) to clarify the example situation depicted in the figure

(A) Probabilistic reversal learning task. Subjects had to choose between a green and a blue stimulus with different reward magnitudes (displayed at the centre of each stimulus). In addition to the reward magnitude, which changed randomly from trial to trial, the value of each stimulus was determined by the probability of reward associated with each stimulus which drifted during the course of the experiment and had to be learned from feedback. After choice, the red bar moved from left to right if the chosen option was rewarded. Subjects tried to reach the silver bar for £10 and the gold bar for £20. **In the example situation depicted, the subject had chosen the green stimulus (indicated by the grey frame on second panel), but was not rewarded so the red bar did not move further towards the right.**

I'm a bit confused by the “signal strength hypothesis”. It is described as the case where the value estimates remain constant but the signal strength of some voxels increase. What do the authors hypothesize is changing physiologically? More neurons with a particular selectivity are recruited? Neurons change their firing rates to a greater degree?

As the reviewer acknowledges, these are only hypotheses where we have to remain agnostic about the precise physiological changes that might lead to the voxel-wise BOLD signal readouts. However, we have been thinking along the same lines as the reviewer is suggesting, an increased signal strength (or signal-to-noise ratio) might be due to more neurons reflecting the information (possibly due to reverberant local circuit activity) or the same number of neurons reflecting the information but with less noise (i.e. change in firing rates), or a combination of both mechanisms. These changes would align with recurrent neural network accounts of information processing with mutual inhibition (e.g. Hunt & Hayden, NatRevNeurosc., 2017) that argue that such local networks can maintain information about previous states and serve as substrates for decision making. We have clarified that these are two possible mechanistic explanations for the effects that we see in the revised Discussion as follows:

The dynamic increase in signal strength (or signal-to-noise ratio) in voxels representing behaviourally relevant LRs in the ACC might be due to more neurons reflecting the information (possibly due to reverberant local circuit activity) or due to the same number of neurons reflecting the information but with less noise (i.e. change in firing rates), or a combination of both mechanisms. These changes would align with recurrent neural network accounts of information processing with mutual inhibition⁴³ that argue that such local networks can maintain information about previous states and serve as substrates for decision making.

In the figures, it would be helpful to label the color bars/axes according to the actual LRs (0 to 1), since that has a direct interpretation in the data, rather the identifying number (1 to 30), which is less intuitive

We have adapted the figures as requested.

Reviewer #3 (Remarks to the Author):

Meder and colleagues report results showing topographical encoding of option values with different learning rates (LR) in the dorsal anterior cingulate cortex (dACC) and inferior parietal lobule (IPL) in humans performing a probability shifting task with different levels of volatility as in Behrens and colleagues 2007. The authors performed a computational analysis of brain activity independently of model fitting to the behavior. Such a modeling approach allows for the search of neural correlates of different model parameters (here LR) which are not necessarily reflected in the subjects' behavior.

The authors found evidence for multiple values with different time constants (i.e. putatively learned with different LRs) within the decision-aligned activity of dACC and IPL. They moreover found that the specific LR encoded by a particular voxel within these regions would increase under high volatility and decrease under low volatility. In addition, they report an increase specific to dACC in the correlation between the best-fitting LRs and their beta-weights under high volatility compared to low volatility. In contrast, vmPFC showed no shift in the distribution of LRs depending on the volatility of the environment. Finally, focusing on the residuals of the previous correlations, they found that voxels in regions such as dACC, IPL and frontopolar cortex preferentially interacted with voxels with similar LRs in other brain regions within this set. In contrast, vmPFC or VS did not show such a selectivity, illustrating that this is not a general tendency of interaction between brain regions nor due to noise but a specific effect to multiple value encoding regions.

The approach and results are very interesting but some aspects of the methods may not be appropriately controlled.

We thank the reviewer for their interest in the manuscript. We have performed the additional analyses requested and have set out a summary of their results below.

Major Points

1. I found slightly unfair the computation of exceedance probability of a GLM with regressors from all 30 LRs plus singular values from SVD compared with a GLM with the absence of any LR regressors. In the latter GLM, why not also performing SVD based on the basic regressors and including the singular values? Could the authors also give the exceedance probability of this comparison when the first GLM does not contain SVD singular values? In addition to comparing multiple LRs with no LRs, I think the authors should also show evidence for multiple LRs against a single best LR. Why not then also testing alternative GLMs such as one containing a single LR which best fits the behavior of the subjects? And another GLM using a single LR explaining the highest variance of dACC activity? Could the authors show exceedance probability for each comparison of pairs of GLMs but also for the global comparison of all above mentioned GLMs? Finally, did the author include the prior over model parameters in the computation of the exceedance probability (which should be computed based on Bayesian evidence rather than likelihood)?

We have performed additional model comparisons along the lines suggested by the reviewer. However, we first note that there may have been a slight misunderstanding about the analysis that we did carry out. The model comparison that we used compared a model with singular values from the SVD with a model without singular values from an SVD. Thus the first model did not have

additional regressors from all 30 LRs. Had we done that then there would have been, of course, a risk that a model containing so many best fitting LRs was over-fitted to the data and so it would have been difficult to compare it against a different model. However, we can avoid this risk by comparing a model with singular values from the SVD with a model without such singular values. This approach allows us to conduct a more general test of whether LR-related information helps explain variance in BOLD and avoids the risk of “double dipping” and overfitting. This is what we presented in the first version of our manuscript. Below we remind the reviewer what that result looked like. On the left (panel A) are the regions of interest and on the right (panel B) are the voxels in which there was more evidence for a multiple LR rate model:

In addition, however, we have followed some of the reviewer’s suggestions: a comparison between a model with the SVD singular values and a model using the single best LR. We have done this in two ways. First we used the LR that would lead to optimal choice behaviour from an optimal Bayesian observer model (see Behrens et al. 2007 for details). Second we used the LR that was fit best to behaviour. Thus, in summary, we have performed the following two complementary analyses, comparing the model with singular values against:

- 1) the LR that generates optimal choice-behaviour (maximizes the reward) against the SVD-model
- 2) the LR that best fits behaviour against the SVD-model.

We show below that when we compare our multiple LR model against a model using optimal behavioural LRs (new Figure S2A, below) or against a model using the LR fit to behaviour (new Figure S2B, below), then the approach yields a similar (albeit slightly weaker) result to that which we showed previously in the same ROI (which we note was defined by independent criteria to avoid any circularity of analysis). In both A) and B), the top row shows the protected exceedance probability in favour of the singular value model at a level of $p > 0.95$, the bottom row at a level of $p > 0.50$.

Methodological details:

For the first analysis, we asked which choice pattern based on the choice probabilities generated by the different LRs would lead to optimal (gain-maximizing) choice behaviour. We then used these LRs to generate choice probabilities and added that regressor to the design matrix, comparing this model against the model with the three singular value regressors in the same way we had compared it against a model without LR regressors reported in the main manuscript.

For the second analysis, we used a hierarchical Bayesian model fitting procedure (Stan (Carpenter et al. (2017) J Stat Softw), mc-stan.org, with Matlab interface) in order to fit a standard reinforcement learning model to the subjects' behaviour. Since the group level average parameter fit has been shown to give the best power for group level fMRI (Ahn et al. (2011) J Neurosc Psychol

Econ), we used this learning rate of 0.25 to generate an LR regressor to compare against the model with the three singular value regressors.

We have added this information to the supplementary material.

Concerning the final question, we did not use log-likelihood scores but AIC scores as an approximation for the log-evidence. These scores were then provided to the `spm_bms` function in SPM 12.

2. The authors claim using a ‘new approach’ to analyse neural data with computational models independently of behavior fitting. However, other groups have already performed similar approaches which should be cited here: Lee, S. W. W., Shimojo, S. & O’Doherty, J. P. Neural Computations Underlying Arbitration between Model-Based and Model-free Learning. *Neuron* 81, 687–699 (2014). Wilson, R. C. & Niv, Y. Is model fitting necessary for model-based fMRI?. *PLoS Comput Biol* 11.6: e1004237 (2015). I think that other groups have subsequently used this method such as the group of Bernard Balleine.

We agree that other groups have analysed neural data with computational models independently of behaviour fitting. However, the simultaneous topographic mapping approach that we follow in this manuscript is something novel. In the revised manuscript we now acknowledge the work the reviewer cites towards the end of the Introduction but also note how our work is different:

Previous investigations have considered neural correlates of model parameters fit to models that do not correspond to the current behavior^{e.g. 6,7} and the issue of the similarity of neural correlates of models with different parameterizations.⁸ However, here we aim to reveal the dynamic changes in and topography of “hidden” information by fitting LR values to each voxel independently, visualising those parameters over anatomical space and computing their interactions.

3. Did the authors use corrections for multiple comparisons when searching for correlations between voxels and the 30 different LRs? Especially when looking for a gradient within dACC?

We did perform multiple comparisons where necessary. As we report in the manuscript, when testing for gradients we a) ran the permutation tests for a general combination of x,y,z-coordinates to predict best-fitting LR and b) only tested for the y-direction in the ACC since this is the only coordinate direction that corresponds (to a certain degree) to the main axis of anatomical organization in the ACC. Since no major direction of anatomical direction has been reported for the IPL, we did not have a similar a priori reason for conducting a similar test for any direction in IPL.

4. The current formulation is sometimes unclear as to whether the authors used as regressors the probability of the chosen option and the prediction error (PE) from a classical reinforcement learning model or from the Bayesian learner model of Behrens and colleagues (2007). The ambiguity lies on the fact that the authors seem to alternate between analyses with the two models. Although the methods are sometimes clearer (stating that the probability of chosen option and PE used in the design matrix with 30 different LRs come from a reinforcement learning model), I think it would help the reader to make it clearer throughout the manuscript. For instance, when presenting

the results in Figure 2. Also when stating that the authors did not replicate the results of Behrens and colleagues (2007) about PE encoding in the dACC. Could this be due to the authors using a reinforcement learning model rather than the Bayesian learner? Since the authors use both models here, could they please make sure they replicate the results of Behrens and colleagues (2007) with the Bayesian learner, and if not, could they discuss and interpret it?

We regret that we did not convey clearly enough that for the entire analysis (with one small exception, see below), we only used a reinforcement learning model and not the Bayesian learning model. The Bayesian learning model was only mentioned/used in two places: 1) for conveying the experimental structure (and the resulting changes in volatility levels) in figure 2 and 2) for the GLM used to retrieve the beta-weight of the regressor coding for the magnitude of the chosen option's potential reward. We now try to clarify this by

- *adding the second sentence below to the "Experimental Strategy" section:*

In the previous study Behrens et al.¹ assumed one dynamic, but unitary LR generating value estimates across the brain. However, in this study we instead compared value estimates generated by 30 stable learning rates.

- *adding the following sentence to the figure legend of figure 2:*

Note, the reward rate and volatility estimates from the Bayesian learner are only shown to convey task structure as well as the difference in volatility between sub-sessions. The Bayesian learner model was not used for analysis.

- *adding the following sentence to the Methods section where we describe the GLM for retrieving the beta-weight for the magnitude of the chosen option:*

Note that this is the only GLM where the Bayesian learning model was used, and it was only used in order to include value estimates while not having to arbitrarily choose one of the 30 LR-regressors from the reinforcement learning model.

We now understand that our statement about the LR-based representation at decision outcome could easily be misunderstood (final sentence before discussion "While Behrens et al.¹ found an overall change in dACC activity during outcome, there was no evidence in the current study for a prediction error signal in dACC, using either standard analysis procedures similar to those used before¹ nor based on Bayesian group model comparisons such as those employed here.").

What we attempted to say was that the previous study had found a main effect of the outcome phase on dACC activity (i.e. there was a general increase in dACC during outcome compared to baseline). This effect we can replicate. However, as in our current analysis, in the previous study there was no effect of PE seen in the dACC (see supplementary figure 2b in Behrens et al. (2007)).

We now rephrased the sentence to say:

As in the study by Behrens et al.¹, in the current study there was an overall change in dACC activity during outcome but no evidence for a prediction error signal in dACC, using either standard analysis procedure similar to those used before¹ nor based on Bayesian group model comparisons such as those employed here.

5. How circular is the calibration of the gap between LRs as a function of the correlation between these regressors with respect to each subject's brain activity? Could the authors instead a priori calibrate the gap between LRs in order to equalize the correlation between the model timeseries generated by model simulation independently of subjects' behavior and brain activity?

We agree with the reviewer that there might be a problem with circularity if the correlation analysis was related to brain activity or if they were in some way fit to behaviour. However, as we state in the experimental procedures, the resulting 30 regressors are based on the 99 LR regressors (not on any BOLD time-course), and are thus unrelated to any brain activity. Since the regressors are modelling the probability of the chosen option, they are necessarily related to behaviour (since their choice defines whether the regressor uses the value of the green or the blue option), but it is not fit to behaviour.

Since reward magnitudes are changing unpredictably, participants estimate reward probabilities and not action values. Thus, for each subject, we then calculated the probability estimates for each stimulus from a simple reinforcement learning model⁵⁵, based on all 99 LRs (α) between 0.01 and 0.99. The model estimates the probability of one of the stimuli leading to a reward by updating the stimulus-reward probability $p(a)$ with LR α , where $R = 1$ when the stimulus was rewarded and $R = 0$ if not:

$$p(a_i) = p(a_{i-1}) + \alpha[R - p(a_{i-1})]$$

The probability estimate of the other stimulus $p(B)$ is $1 - p(A)$. (...) After normalising the probability estimates for all LRs for both stimuli, we derived the probability estimate of the chosen stimulus $p(\text{chosen})$. These $p(\text{chosen})$ -regressors (hereafter “LR regressors”) and the PE regressors were convolved with the HRF, normalised and high-pass filtered in the same way (in the same manner as in FSL). We calculated a correlation matrix for the 99 resulting LR regressors for every subject and for the whole session as well as the two sub-sessions. Since the correlation between regressors is not the same for all levels of LR, we chose 30 regressors that were equally spaced in terms of their correlation to the neighbouring regressors. We did so by averaging the 30 LR regressors with equal correlation for every subject in all three sessions and subsequently rounding them to two decimals. This procedure resulted in 30 LR regressors corresponding to the following LRs (see also Fig. 2):
[0.01 0.02 0.03 0.04 0.05 0.06 0.07 0.08 0.09 0.11 0.12 0.14 0.15 0.17 0.20 0.22 0.25 0.28 0.32 0.36 0.40 0.46 0.51 0.57 0.64 0.71 0.78 0.85 0.93 0.99].

6. Figure 8B: could the authors interpret why the matrix contains high correlation values in the borders as well as in horizontal line #18 (dACC) and vertical lines #7 and #19 (IPL)?

We are not sure that such effects are significant and given the care we have taken in avoiding multiple comparisons we are reluctant to run such tests. We think that these are likely to be minor effects in the data.

7. Line 426, the authors should show the stats (incl. exceedance probability) when referring to the results of the analysis for the vmPFC.

We have added the following information:

Unlike in dACC and IPL, in vmPFC the amount of BOLD variance explained by SVD-derived singular values reflecting the LR regressors was not significantly greater than the amount of variance explained by a model lacking LR information (mean protected exceedance probability in

anterior vmPFC = 0.478, t-test against 0.5: $t_{1007} = -5.19$, $p < 0.001$. Mean protected exceedance probability in posterior vmPFC = 0.340, $t_{1241} = -33.274$, $p < 0.001$).

8. I found very interesting and important the analysis showing interaction between voxels of different regions with similar LRs in contrast to regions which do not show such an effect. Since such an effect may reflect parallel estimations which at some point need to converge/integrate in order to make a decision, I am wondering whether the authors could push further the analysis to identify potential regions where the analysis would suggest a convergence/integration. This would definitely strengthen the whole demonstration.

We thank the author for the suggestion. While we think that this is a very good idea, it would require elaborate considerations on how to perform this analysis and time for its implementation. We would therefore leave this analysis to subsequent projects.

9. Also in terms of interpretation, could the authors discuss their results with respect with other types of gradients reported in terms of reinforcement learning models? Tanaka, S. C., Doya, K., Okada, G., Ueda, K., Okamoto, Y., & Yamawaki, S. Prediction of immediate and future rewards differentially recruits cortico-basal ganglia loops. *Nature neuroscience*, 7(8), 887-893 (2004). Holroyd, C. B., & McClure, S. M. Hierarchical control over effortful behavior by rodent medial frontal cortex: A computational model. *Psychological review*, 122(1), 54 (2015).

We thank the author for the suggestions. We have added the following sentences referring to the study by Tanaka et al. in the discussion:

We are aware of one other study showing the representation of multiple estimates for latent decision-variables, in that case of reward prediction errors based on different discount factors.⁷ However, here we tried to fully exploit the potential of topographic representations by showing their dynamic adaptations as well as interregional connectivity patterns.

Since the paper by Holroyd et al. seems to make a strong case about hierarchical gradients between different regions (rather than within regions), we added a reference to their paper to our discussion of other studies showing and discussing such hierarchies of brain regions:

The evidence for value learning using multiple LRs in several cortical areas fits well with the idea that there exists a hierarchy of information accumulation from short time scales in sensory areas to long time scales in prefrontal, dACC, and parietal association areas⁴³⁻⁴⁸.

Minor Points

We thank the reviewer for the careful reading and have corrected the mistakes/improved the wording as suggested in points 1, 2, 5 and 6.

1. Since there is no value update hypothesized by reinforcement learning models at the decision time, to make the sentence in line 52 clearer, the authors could reformulate in the following manner:

'It is therefore unclear how dACC activity at the point in time when decisions are actually made change as a function of etc.'

Changed as requested.

2. Along the same lines, line 69: 'calculate a single value' -> 'consider a single previously calculated value'.

Changed as requested.

3. What are the links and differences between Figure 4A and Figure 2E? Do they come from the same analysis? This is not clear to the reader.

Yes, panel 4a and 2e show the same data. Panel 4a shows it as part of an illustration of the analysis steps. Panel 4a shows it in more details next to example subject data so that the result can be examined more clearly.

4. Figure 7B: why not trying a linear regression and if it is significant, showing that the slope is negative in the upper panel and positive in the lower one?

We have done something very similar to the reviewer's suggestion in panel A of figure 7. In panel A we show that correlation coefficients for the data shown in the upper and lower parts of 7b and show that the correlation coefficients are significantly different.

5. Line 696: enssure -> ensure.

Changed as requested.

6. Supplementary material lines 158 and 160: Fig. S8A -> Fig. S6A and Fig. S8B -> Fig. S6B.

Changed as requested.

REVIEWERS' COMMENTS:

Reviewer #2 (Remarks to the Author):

I have reviewed the revised manuscript, and the authors have adequately addressed the concerns I raised. I think this is an interesting study that makes a substantive contribution to the field.

Reviewer #3 (Remarks to the Author):

The authors have addressed all my concerns.